# Marine Phytoplankton Stoichiometry Mediates Nonlinear Interactions Between Nutrient Supply, Temperature, and Atmospheric CO$_2$

Allison R. Moreno[1], George I. Hagstrom[2], Francois W. Primeau[3], Simon A. Levin[2], Adam C.
Martiny[1,3*]

Affiliations:
1. Department of Ecology and Evolutionary Biology, University of California, Irvine, California 92697, USA.
2. Department of Ecology and Evolutionary Biology, Princeton University, Princeton, New Jersey 08544, USA.
3. Department of Earth System Science, University of California, Irvine, California 92697, USA.

*Corresponding Author
amartiny@uci.edu

Keywords: Redfield Ratio, Traits, Carbon Cycling

**Abstract**

Marine phytoplankton stoichiometry links nutrient supply to marine carbon export. Deviations of phytoplankton stoichiometry from Redfield proportions (106C:1P) could therefore have a significant impact on carbon cycling, and understanding which
environmental factors drive these deviations may reveal new mechanisms regulating the carbon cycle. To explore the links between environmental conditions, stoichiometry, and carbon cycling, we compared four different models of phytoplankton C:P: a fixed Redfield model, a model with C:P given as a function of surface phosphorus concentration ([P]), a model with C:P given as a function of temperature, and a new multi-environmental model
that predicts C:P as a function of light, temperature, and [P]. These stoichiometric models were embedded into a five ocean circulation box model, which resolves the three major ocean biomes (high-latitude, subtropical gyres, and tropical upwelling regions). Contrary to the expectation of a monotonic relationship between surface nutrient drawdown and carbon export, we found that lateral nutrient transport from lower C:P tropical waters to
high C:P subtropical waters could cause carbon export to decrease with increased tropical nutrient utilization. It has been hypothesized that a positive feedback between temperature and $pCO_{2,atm}$ will play an important role in anthropogenic climate change, with changes in the biological pump playing at most a secondary role. Here we show that environmentally driven shifts in stoichiometry make the biological pump more influential, and may reverse
the expected positive relationship between temperature and $pCO_{2,atm}$. In the temperature-only model, changes in tropical temperature have more impact on the $\Delta pCO_{2,atm}$ ($\sim$41 ppm) compared to subtropical temperature changes ($\sim$4.5 ppm). Our multi-environmental model predicted a decline in $pCO_{2,atm}$ of $\sim$46 ppm when temperature spanned a change of 10°C. Thus, we find that variation in marine phytoplankton stoichiometry and its
environmental controlling factors can lead to non-linear controls on $pCO_{2,atm}$, suggesting the need for further studies of ocean C:P and the impact on ocean carbon cycling.

## 1 Introduction

The discovery of large-scale deviations of phytoplankton stoichiometry from the Redfield ratio in the past decade (Martiny et al., 2013a, 2013b; Weber and Deutsch, 2010) has significant consequences for our understanding of the biological carbon pump and global carbon cycling (Galbraith and Martiny, 2015; Moreno and Martiny, 2018). Traditionally, the biological pump is thought to be controlled by a combination of the vertical nutrient flux

and nutrient utilization efficiency (Sarmiento and Toggweiler, 1984). Evidence that elemental stoichiometry is variable thus adds a new dimension to the biological pump, and may lead to higher than currently expected carbon export in subtropical regions (Emerson, 2014; Tanioka and Matsumoto, 2017; Teng et al., 2014). Global carbon export has been estimated to range between 5 and 12 Pg C/year (Boyd and Trull, 2007; Henson et al.,

2011), but these projections have yet to incorporate the environmental controls on $C:P_{export}$. Variation in $C:P_{export}$ from Redfield proportions can be linked to environmental conditions. There are two leading environmental parameters thought to control $C:P_{export}$; nutrients, predominantly phosphate concentrations, and temperature. Galbraith and Martiny used a simple three-box model to show that variable stoichiometry driven by

phosphate availability could enhance the efficiency of the biological pump in the low-latitude ocean (Galbraith and Martiny, 2015). In contrast, Yvon-Durocher and co-workers (2015) used a meta-analysis of global temperature and stoichiometric ratios to propose that C:P increased 2.6-fold from 0° C to 30° C. Thus, it is unclear if differences in nutrient supply, temperature, or some combination of them, control the global variation in C:P of

plankton and exported material.

There are two important ingredients missing from published studies that could alter the interactions among phytoplankton stoichiometry, carbon export, and atmospheric $pCO_2$ ($pCO_{2,atm}$). The first is the presence of two distinct low-latitude biomes, namely the equatorial upwelling regions and the macronutrient-depleted subtropical gyres. In direct

observations and inverse model analyses, these two biome types appear to have unique elemental compositions (DeVries and Deutsch, 2014; Martiny et al., 2013a; Teng et al., 2014). Thus, in order to properly represent global variations in surface plankton C:P and carbon export, it is essential to separately model macronutrient-limited subtropical gyres and equatorial upwelling zones.

The second missing ingredient is that environmental factors beyond nutrient availability may impact the elemental composition of surface plankton and $C:P_{export}$. Temperature, irradiance, and nutrient concentrations are all important environmental factors, which influence the physiology and stoichiometry of phytoplankton. However, surveys of phytoplankton C:P are insufficient to distinguish the separate effects of each

factor on C:P due to strong environmental covariance. Cellular trait-based models use detailed studies of phytoplankton physiology to determine how phytoplankton cells should allocate their resources as a function of environmental conditions, allowing us to model the interactive influence of temperature, nutrient concentrations, and irradiance on C:P ratios (Clark et al., 2011; Daines et al., 2014; Shuter, 1979; Talmy et al., 2014; Toseland et al.,

2013). Numerous physiological mechanisms have been proposed to explain variation in phytoplankton stoichiometry, including growth rate (Sterner and Elser, 2002), photoacclimation (Falkowski and LaRoche, 1991; Geider et al., 1996; Leonardos and Geider, 2004, 2005), nutrient-limitation responses (Garcia et al., 2016; Goldman et al., 1979; Rhee, 1978), and temperature acclimation (Rhee and Gotham, 1981; Toseland et al.,

952013; Yvon-Durocher et al., 2015). Through incorporation of such physiological responses, a trait-based model has revealed that differences in ribosomal content and cell size between warm-water, oligotrophic environments and cold-water, eutrophic environments are important mechanisms driving stoichiometric variation in the ocean (Daines et al., 2014). Thus, linking biome-scale variations in environmental conditions with a detailed 100trait-based model of phytoplankton resource allocation and elemental composition may enable us to more fully explore interactions among multiple ocean environmental factors, the biological pump, and $pCO_{2,atm}$.

Here, we create a five ocean circulation box model, incorporating the three major ocean biomes, to study the feedback effects of variable stoichiometry on carbon export and 105$pCO_{2,atm}$. We will explicitly address the following research questions: (1) How does environmental variability influence marine phytoplankton cellular allocation strategies and in turn the elemental stoichiometric ratio? (2) What are the effects of changing environmental conditions on stoichiometric ratios, carbon export, and $pCO_{2,atm}$?, and (3) What is the influence of the environmental conditions among the three major surface 110biomes on carbon export and $pCO_{2,atm}$?

**2 Methods**
**2.1 Box Model Design**
To quantify the feedbacks between phytoplankton stoichiometry, carbon export, and 115$pCO_{2,atm}$, we formulated a five-box ocean circulation model of the phosphorus and carbon cycles in the ocean coupled to an atmospheric box. The foundation of our model is based on the models introduced in Ito and Follows (2003) and DeVries and Primeau (2009). Phosphorus is used to represent the role of nutrient availability in controlling stoichiometry and C export. We chose this over N because on long time-scales, P is 120commonly considered the ultimate limiting nutrient whereas N is only limiting productivity and export on short time-scales (Tyrrell, 1999). On long time-scales, nitrogen fixation/denitrification will presumably adjust the N inventory. Our modeling is focused on long term steady-state outcomes and we would like to avoid issues associated with modeling the N cycle (like getting N-fixation and denitrification rates correct). Thus, we 125chose to use P as a representative for nutrient availability at long-term steady-state biogeochemical equilibrium. The model includes three surface boxes, each corresponding to one of the major biomes: the tropical equatorial upwelling regions (labeled T̲), the subtropical gyres (labeled S̲), and the high-latitude regions (labeled H̲) (Figure 1). We define the oligotrophic subtropical gyre regions where the mean annual phosphate 130concentration is less than 0.3 $\mu$M (Teng et al., 2014), with the remainder of the surface ocean assigned either to box T̲ or box H̲ based on latitude. We use these assignments to calculate the baseline physical properties of each region, including mean annual averaged irradiance and temperature. The subsurface ocean is divided into two regions: the thermocline waters that underlies the subtropical gyres and  the equatorial upwelling 135regions (labeled M̲), and deep waters (labeled D̲) (DeVries and Primeau, 2009).

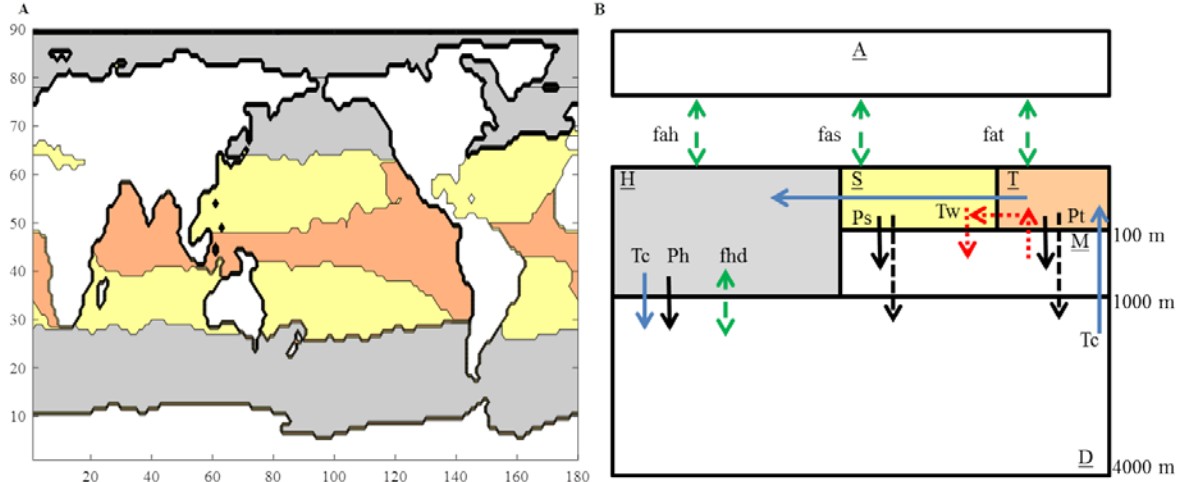

**Figure 1: Box Model Design**. *A) Sea surface breakdown by region. All peach-colored regions represent the tropical surface ocean box, the cream-colored regions represent the subtropical surface ocean box, and grey regions represent the high-latitude surface ocean box. B) The model includes tropical (T), subtropical (S), and high-latitude (H) surface ocean boxes, a mixed thermocline (M) box, and a deep water (D) box. The thermohaline circulation Tc is set to 20 Sv, while the wind driven shallow overturning circulation is set to 5 Sv. The high-latitude mixing flux fhd is set to 45.6 Sv. The thickness of Box H is 1000 m, and Box M is 900 m. Box T has a temperature of 26°C, box S has a temperature of 24°C, and box H has a temperature of 7°C. Box S covers 39% and Box T covers 25% of the ocean surface area.*

To simulate the global transport of water between boxes, our model includes a thermohaline circulation (Tc) that upwells water from the deep ocean into the tropics, laterally transports water into the subtropics and high-latitudes, and downwells water from the high-latitude region to the deep ocean. Surface winds produce a shallow overturning circulation (Tw), that transports water from the thermocline to the tropics and then laterally into the subtropics. These circulations create teleconnections of nutrient supply in the surface ocean boxes. A bidirectional mixing term that ventilates the deep box directly through the high-latitude surface box (fhd) represents deep water formation in the Northern Atlantic region and around Antarctica (Sarmiento and Toggweiler, 1984). The parameters Tc, Tw and fhd are considered adjustable parameters, which we calibrate using phosphate data from WOA13 (Garcia et al., 2014). In order to simulate the movement of particles, we included export fluxes (Pt, Ps, and Ph) of organic phosphorus out of each surface box. The conservation equations of phosphorus are as follows:

$$\frac{dP_T}{dt} = \frac{(P_M - P_T) \cdot Tc + (P_M - P_T) \cdot Tw - (a+b) \cdot Pt}{VT} \qquad (1)$$

$$\frac{dP_S}{dt} = \frac{(P_T - P_S) \cdot Tc + (P_T - P_S) \cdot Tw - (a+b) \cdot Ps}{VS} \qquad (2)$$

$$\frac{dP_H}{dt} = \frac{(P_S - P_H) \cdot Tc + (P_D - P_H) \cdot fhd - Ph}{VH} \qquad (3)$$

$$\frac{dP_M}{dt} = \frac{(P_D - P_M) \cdot Tc + (P_S - P_M) \cdot Tw + a \cdot Pt + a \cdot Ps}{VM} \qquad (4)$$

$$\frac{dP_D}{dt} = \frac{(P_H - P_D) \cdot Tc + (P_H - P_D) \cdot fhd + Ph + b \cdot Pt + b \cdot Ps}{VD} \qquad (5)$$


where P represents the concentration of phosphorus at a specific box, a represents 0.25 remineralization, b represents 0.75 remineralization, and V represents the volume of the specified box.

Our box model simulates [P], alkalinity and various forms of C; total carbon in the
surface boxes is partitioned into carbonate, bicarbonate, and $pCO_2$. The global mean [P] is prescribed according to the observed mean value (Garcia et al., 2014). The export of carbon is linked to phosphorus export using the C:P$_{export}$ ratio. To quantify the breakdown of carbon into these components, we model the solubility pump, using temperature and salinity to determine the partitioning of inorganic carbon among total carbon within a box.
The global mean alkalinity is prescribed according to the observed mean ocean values but is also subject to transport (Sarmiento and Toggweiler, 1984). Our box model simulates alkalinity and total inorganic carbon, which are conserved tracers from which the speciation of inorganic carbon in seawater can be calculated. Biome specific salinity and temperature are used to prescribe the solubility constants of $CO_2$ in seawater and the
bromine concentration, which is taken to be proportional to salinity. $CO_2$ cycles through the atmosphere via the air-sea gas exchange fluxes (fah, fas, fat). We used a uniform piston velocity of $5.5 \times 10^{-5}$ m s$^{-1}$ to drive air-sea gas exchange (DeVries and Primeau, 2009; Follows et al., 2002). Quantifying the atmospheric concentration of carbon satisfies:

$$\frac{dC_A}{dt} = [(C_T - C_A) \cdot SolT(temp, sal) \cdot fat + (C_S - C_A) \cdot SolS(temp, sal) \cdot fas +$$
$$(C_H - C_A) \cdot SolH(temp, sal) \cdot fah]/VA \qquad (6)$$

where C represents the concentration of total carbon in a specific box , Sol is the solubility constant in a specified box, calculated from temperature (temp) and salinity (sal).

We calibrated our model parameters (Tc, Tw, fhd) so that the macronutrients were
at similar average values compared to World Ocean Atlas 2013 dataset for each location. We tested the sensitivity of modeled $pCO_{2,atm}$ to the fluxes Tc, Tw, and fhd and found that with Tc = 20 Sv and Tw = 5 Sv (values that allowed the model to match [P] and alkalinity), $pCO_{2,atm}$ was sensitive to the value of fhd (Sarmiento and Toggweiler, 1984). Guided by values previously used in the literature, we set fhd to 45.6 Sv (Table 1) but we also present
results for the nutrient-only stoichiometry model at two extreme values of fhd (18 and 108 Sv) (Figure 2). The functional dependence of $pCO_{2,atm}$ with changing subtropical and tropical [P] for each extreme value of fhd was quite similar, though the value of $pCO_{2,atm}$ for the high fhd simulation was approximately twice that of the low fhd simulation (Figure 2). We found that our value of 45.6 Sv provides a modern $pCO_{2,atm}$ value. Although the focus of
this study is to determine the impact of low latitude biogeochemistry on $pCO_{2,atm}$, we point out that at Redfield stoichiometry, $pCO_{2,atm}$ increases by 100 ppm when fhd increased from its default value 45.6 Sv to 108 Sv. For certain values of the parameters, the model produced excessive nutrient trapping in the thermocline. In order to dampen the nutrient trapping, we tuned the remineralization depth. As such, 25% of the total export is respired

210in the thermocline with the remaining 75% exported into the deep ocean, leading to a
better match between the modeled and observed [P] in the thermocline box.

**Table 1: High-latitude deep water exchange range**

| RANGE OF FHD [SV] | SOURCESOURCE |
|---|---|
| 38.1 | (Sarmiento and Toggweiler, 1984) |
| 3-300 | (Toggweiler, 1999) |
| 60 | (DeVries and Primeau, 2009) |
| 30-130 | (Galbraith and Martiny, 2015) |
| 18-108 (default value 45.6) | This Study |


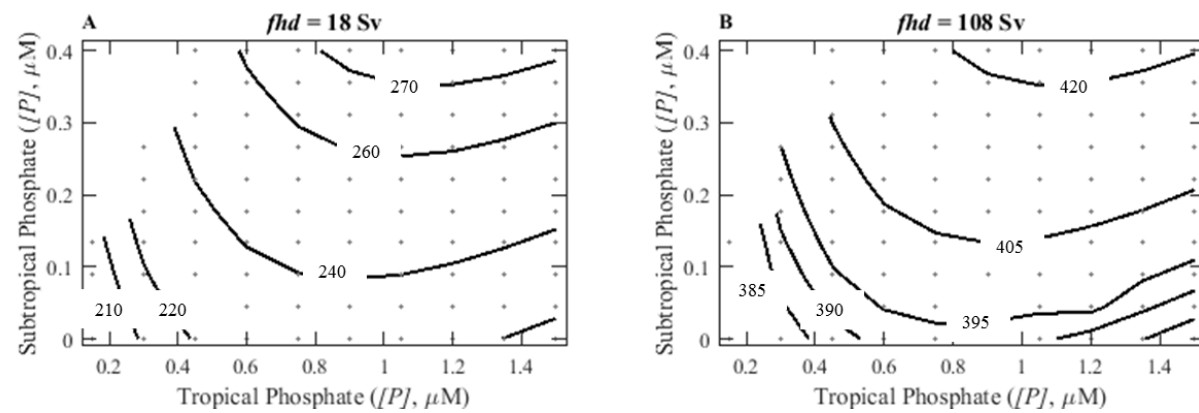

*Figure 2: pCO$_{2,atm}$ (ppm) sensitivity to extreme fhd values under changing surface phosphate
concentrations. A.) Range of pCO$_{2,atm}$ (ppm) using an fhd value of 18 Sv. B.) Range of pCO$_{2,atm}$ (ppm) using an
fhd value of 108 Sv.*

## 2.2 Stoichiometric Models

To quantify and understand the feedbacks between carbon export and pCO$_{2,atm}$, we
embedded four stoichiometric models into our five-box ocean circulation model. Each
225model differs according to its complexity and how much environmental information they
utilize. These are a static Redfield model that assumes that C:P$_{export}$ is constant across
environmental conditions, a nutrient-only model that uses surface [P] to predict C:P$_{export}$
(from Galbraith and Martiny, 2015), a temperature-only model that uses $T$ to predict
C:P$_{export}$ (modified from Yvon-Durocher et al., 2015, and a multi-environmental model that
230uses light, $T$, and [P] to predict C:P$_{export.}$

### 2.2.1 Static Redfield Model

Our control model uses a static Redfield stoichiometry. The Redfield ratio is based on an average value of organic carbon to phosphorus of 106:1.


### 2.2.2 Nutrient-Only Model

The nutrient-only stoichiometric model expresses phytoplankton C:P as a function of the ambient phosphate concentration:

$$C:P = \frac{1}{\kappa[P] + [P]_0} \tag{7}$$

where the parameters $\kappa = 6.9x10^{-3}\mu M^{-1}$ and $[P]_0 = 6.0x10^{-3}$ were obtained by regressing 240 the reciprocal of C:P onto [P] (Galbraith and Martiny, 2015).

### 2.2.3 Temperature-Only Model

The temperature-only stoichiometric model expresses phytoplankton C:P as a function of temperature:

$$ln(C:P) = \Pi(T - 15°C) + b, \tag{8}$$

where the parameters $\Pi = 0.037/°C$ and $b = 5.5938$ (Yvon-Durocher et al., 2015). The temperature-only model was created to determine the temperature responses of log-transformed C:P ratios centered at 15°C.

### 2.2.4 Multi-Environmental Model

We created a multi-environmental model which predicts how cell size, biomass allocations to biosynthesis and photosynthesis, and C:P ratios vary with temperature, light levels, temperature, and phosphorus concentrations. The multi-environmental factor model was derived from a non-dynamic physiological trait-based model. We used a theoretical cellular-allocation trait model based on phytoplankton physiological properties that

divides the 'cell' into several functional pools which represent cellular investments in biosynthesis, photosynthesis, and structure, and a storage pool, which represents variations in the level of P-rich molecules such as polyphosphates (full model equations can be found on GitHub: https://github.com/georgehagstrom/-bg-2017-367-/blob/master/CP.m). The functional pools are composed of biological macromolecules

such as ribosomes, proteins, carbohydrates and lipids. The model predicts the size of each pool as a function of light, $T$, and [P]. The size of each functional pool is modeled by using subcellular resource compartments, which connect the fitness of a hypothetical phytoplankton cell in a given environment to its cellular radius and the relative allocation of cellular material to photosynthetic proteins, ribosomes, and biosynthetic proteins. We

assume that real phytoplankton populations have physiological behaviors that cluster around the strategy that produces the fastest growth rate in each environment (Norberg et al., 2001), and use the stoichiometry of this optimal strategy to model the elemental composition of cellular material (Figure 1).

Phytoplankton can accumulate large reserves of nutrients that are not immediately

incorporated into the functional components of the cell (Diaz et al., 2016; Mino et al., 1998; Van Mooy and Devol, 2008; Mouginot et al., 2015). This storage capability varies among phytoplankton species, and depends on the particular nutrient under consideration: the

cost for storing physiologically relevant quantities of nutrients is low for nutrients with low quotas such as phosphorus, in comparison to nitrogen and carbon. Thus, the phosphorus
storage is assumed highly plastic in comparison to carbon storage (Moore et al., 2013). Further, we assume that each cell dedicates a fixed fraction of its biomass to carbon reserves, and focus our modeling efforts on the variability of the stored phosphorus pool. To predict the size of the storage pool, we assume a linear relationship between stored phosphorus and ambient environmental phosphorus levels and used statistical modeling of
an oceanic C:P dataset (Martiny et al., 2014) to calculate the constant of proportionality. The result is a relatively simple model that both qualitatively and quantitatively predicts the variation of C:P in phytoplankton.

Phytoplankton physiology is modeled through allocations of cell dry mass to three distinct pools: structure ($S(r)$), biosynthesis ($E$), and photosynthesis ($L$). Allocations satisfy:

$$1 = S(r) + E + L,$$ 
(9)

where the variables $S, E$, and $L$ represent the *specific* allocations of cellular biomass.

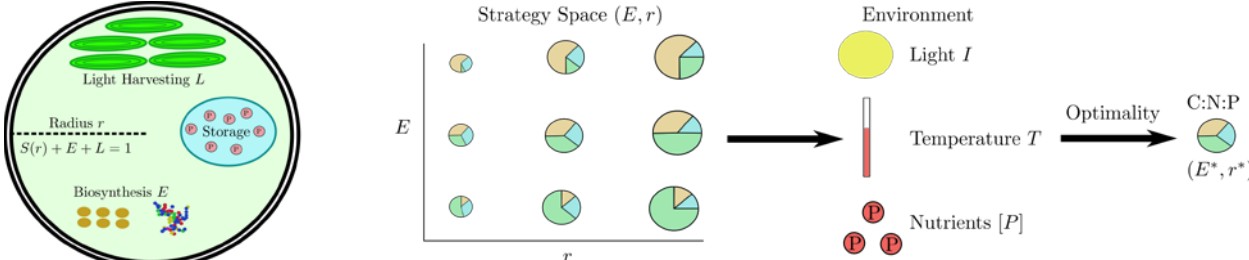

**Figure 3:** *Diagram of physiological model. Phytoplankton strategies are represented in a two-dimensional strategy space (E, r). Each strategy is assigned a fitness in each environment using physiological principles, and*
*the strategy with the highest fitness is selected to represent the local population. The stoichiometry of cellular components is used to calculate the stoichiometry of the functional pools in the cell.*

The specific allocation of biomass to the cell membrane is inversely proportional to the cell radius $\left(\frac{\alpha}{r}\right)$ (Clark et al., 2011), which accounts for the changing relative volume of
the cell membrane with radius. The structure pool includes the cell membrane plus wall and other components ($\gamma$), which are not related to photosynthesis or biosynthesis and is given by:

$$S(r) = \frac{\alpha}{r} + \gamma.$$ 
(10)

In an environment specified by $T$, $[P]$, and light level ($I$), the growth rate of a cell using a given strategy is the minimum of the following growth rates:

$$\mu = min(\mu_E, \mu_L, \mu_P).$$ 
(11)

Here $\mu_E$ is determined by the specific rate of protein synthesis, $\mu_L$ is determined by the specific rate of carbon fixation, and $\mu_P$ is determined by the specific rate of phosphorus uptake, or:

$$\mu_E = k_E(T)E, \mu_L = \frac{f_P(L,I) - \Phi_M(T)}{1 + \Phi_S}, \mu_P = \frac{1}{Q_P(r,E)} \frac{V_m(r)[P]}{K_P(r) + [P]}. \tag{12}$$

We assume that part of the energy captured by a cell via photosynthesis is used for maintenance ($\Phi_M$), whereas the rest is used to drive the synthesis of new macromolecules ($\Phi_S$), so that a cell growing at rate $\mu_L$ is in energy balance. The efficiency of biosynthesis $k_E$ and the carbon cost of maintenance $\Phi_M$ are functions of $T$, whose dependence is modeled using $Q_{10} = 2.0$ (Van Bogelen and Neidhardt, 1990; Broeze et al., 1978; Shuter, 1979).

Uptake is regulated by a Monod function with kinetic parameters depending on the radius through the allometric scaling relationships derived from measurements of phytoplankton populations (Edwards et al., 2012):

$$V_m(r) = a_P r^{b_P}, K_P(r) = a_K r^{b_K}. \tag{13}$$

This use of allometric scaling relationships departs from the conventions adopted by
Shuter (1979) or Daines et al. (2014), who assume that uptake rates are diffusion-limited.

The phosphorus quota for functional elements of the cell (thus not including any storage) is determined by the allocation to biosynthesis $E$ and the percentage $p_{DNA}$ of cellular dry mass allocated to DNA:

$$Q_{p,biosynthesis}(E,r) = \frac{4}{3}\pi r^3 \rho_{\text{cell}} p_{\text{dry}} \frac{(\alpha_E E P_{\text{rib}} + p_{DNA} P_{DNA})}{31}. \tag{14}$$

Here, we assume that there is no contribution to the functional-apparatus P quota from phospholipids, which instead are merged with storage molecules. This differs from Daines et al. (2014), who assumes that phospholipids occupy 10% of the cell by mass. Phytoplankton can substitute sulfoquinovosdiaglycerol (SQDG) for phospholipids in their cell membranes under low P conditions (Van Mooy et al., 2009). Similarly, P storage
molecules are also regulated by P availability. Thus, we treat phospholipids and P-storage as one pool.

The function $f_P$ is the cellular response to light levels, and is chosen to capture the effects of both electron transport and carbon fixation on photosynthesis, and is closely related to a previous model (Talmy et al., 2013). This prior model included four
compartments: electron transport, carbon fixation, photoprotection, and biosynthesis. It was found that photoprotection allocation was not a large or greatly changing component of their allocations. We therefore do not include this within our model due to its high complexity with little qualitative results. Our biosynthesis was also separately parametrized.
The decomposition of photosynthesis into light harvesting and carbon fixation components is critical, and makes our model predictions agree much better with experiments studying the variations of C:P or N:P ratios with irradiance. Models that do not have this decomposition predict too large a decrease in cellular allocations to photosynthesis at high-light levels. In a two- compartment model, increases in allocations
to carbon fixation cause the overall allocation to light harvesting to have a more mild decrease. The two-compartment treatment also seems more physiologically realistic than a

1-compartment treatment, which only models photosynthetic pigments. Thus, we used the functional forms and parameters that were derived (experimentally) previously for carbon fixation and light harvesting (Talmy et al., 2013).

Our model interprets light harvesting allocation, $L$, as being composed of proteins dedicated to carbon fixation ($F_1$), such as RuBisCO, and proteins dedicated to light harvesting ($F_2$), such as photosynthetic pigments. The rate of photosynthetic carbon fixation is a function of the allocations to each of these, which satisfy $F_1 + F_2 = L$. The relative allocations together determine the overall photosynthetic rate:

$$P_{\max} = \min\left(k_1 F_1, k_2 F_2\right), f_p = P_{\max}\left(1 - exp\left(\frac{-\alpha_{\mathrm{ph}}\phi_M F_2 I}{P_{\max}}\right)\right). \tag{15}$$


For a given $I$ and $L$, there is a pair of values $\left(F_{1,\mathrm{opt}}, F_{2,\mathrm{opt}}\right)$ that maximize the photosynthetic rate $f_p$. We estimate the photosynthetic rate $f_p(L, I)$ under the assumption that cells assume the optimal allocations to carbon fixation and electron transport. Our model departs from the models developed by Shuter (1979) and Daines et al. (2014), which

assume that energy acquisition is a linear function of light levels leading to functional responses linearly proportional to the cellular investment in light harvesting proteins.

We model photosynthesis as having a $Q_{10}$=1, which is consistent with physiological studies going back to Shuter (1979) that suggest that photosynthetic efficiency does not depend on temperature over physiologically relevant ranges. The discrepancy between

photosynthetic and biosynthetic temperature dependence has traditionally been explained by referring to the differences in the chemistry and physics of the two processes. The electron transport chain relies on quantum mechanical processes, which are unaffected by variations in temperature in a physiologically relevant range (Devault, 1980). Required maintenance respiration rates are also modeled as having a $Q_{10}$=2.0 (Devault, 1980).

Required maintenance respiration rates are also modeled as having a $Q_{10}$=2.0. We model the phytoplankton community residing in a given environment by assuming it consists solely of the phytoplankton type using the highest growth rate strategy in that environment. This strategy is found by solving for the values of $r$ and $E$ and that make

$$\mu = \mu_L = \mu_P = \mu_E \tag{16}$$

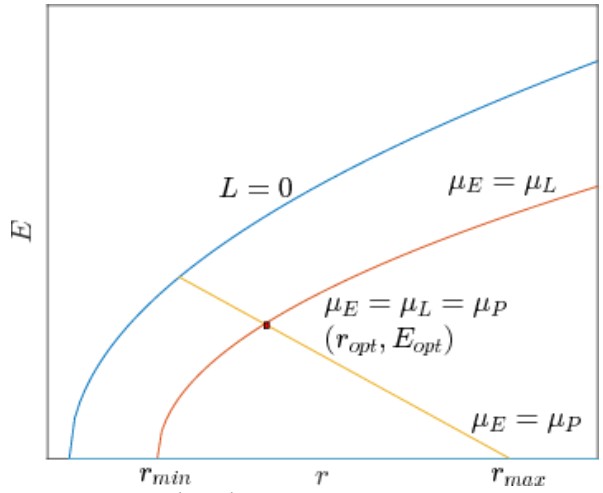


**Figure 4: Diagram of strategy space.** *The $(r, E)$ plane is divided into a region in the first quadrant where $L > 0$, corresponding to the set of allowable strategies. The optimal strategy occurs at the point $(r_{opt}, E_{opt})$, denoted by the red rectangle, where $\mu = \mu_L = \mu_P = \mu_E$.*

We will now show that under two assumptions that will be true in nearly any realistic situation, a strategy maximizing $\mu$ always exists, is unique, and satisfies $\mu = \mu_L = \mu_P = \mu_E$ (Figure 4). The function $\mu_L$ is a function of the chosen strategy $(r, E)$, and it is an increasing function of $r$ and decreasing function of $E$. The first assumption is that light levels are sufficiently high that there exists some $r_{min}$ such that $\mu_L(r_{min}, 0) > 0$, which means that

light is sufficient for some phytoplankton to be able to overcome maintenance costs. The function $\mu_P$ is a monotonically decreasing function of both $r$ and $E$. As there is a non-zero amount of $P$ contained in the structure pool, and because uptake rates decline to zero with $r$, there will be some $r_{max}$ at which $\mu_P(r_{max}, 0) > 0$. The second assumption is that $r_{min} < r_{max}$, which will be true for most realistic values of the light level. We note that for

fixed $r$, $\mu_E$ is a monotonically decreasing function of $E$. Since none of $\mu_E$, $\mu_L$, or $\mu_P$ have critical points, the function $\mu$ can only have a maximum at places where two or more of $\mu_L$, $\mu_P$, and $\mu_E$ are equal, or at the boundaries of the strategy space. On the boundaries of strategy space, $E = 0$ or $L = 0$ so that $\mu \le 0$. We can exclude the boundary and focus on places where two or more of $\mu_L$, $\mu_P$, and $\mu_E$ are equal. We define two curves, one on which

$\mu_L = \mu_E$, and the other on which $\mu_P = \mu_E$. The curve for which $\mu_L = \mu_E$ begins at the point $r = r_{min}$ and can be described by a monotonically increasing function $E = g(r)$ on the interval $[r_{min}, \infty]$. This curve exists because $\mu_E = 0$ when $E = 0$, $\mu_L > 0$ when $E = 0$ and $r_{min} < r$, and $\mu_L < 0$ when $L = 1 - S(r) - E = 0$, so that there is always a solution to $\mu_L = \mu_E$ for fixed $r > r_{min}$. To see that the curve is an increasing function of $r$, consider the

function $V(E, r) = \mu_L - \mu_E$ and apply the chain rule to the equation $V(g(r), r) = 0$ to find that along the curve E=g(r) :

$$\frac{dE}{dr} = g'(r) = \frac{-\dfrac{\partial V}{\partial r}}{\dfrac{\partial V}{\partial E}} \tag{17}$$

We consider the terms in equation 17 carefully. The function $V$ is an increasing function of
$r$ because $\mu_E$ is independent of $r$ and because $\mu_L$ is an increasing function of $r$ (for a fixed investment in biosynthesis, a larger radius leads to a greater investment in photosynthesis and greater photosynthetic growth rate). Thus, the numerator of equation 17 is negative. The function $V$ is a decreasing function of $E$ because $\mu_L$ is a decreasing function of $E$ (greater investments in biosynthesis at fixed radius lead to smaller investments in
photosynthesis) and $\mu_E$ is an increasing function of $E$. Thus the denominator of equation 17 is negative, and the quotient on the right hand side is positive, so $g'(r)$ is positive and describes an increasing curve.

By similar logic, we can define a curve $h(r)$ that solves the equation $\mu_P(h,r) = \mu_E(h,r)$. This curve exists on the finite interval $[r_I, r_{max}]$, where $r_I$ solves the equation
$\mu_P(1 - S(r_I), r_I) = \mu_E(1 - S(r_I), r_I)$. Thus, $h(r)$ represents a decrasing curve from the point $(1 - S(r_I), r_I)$ to $(0, r_{max})$. We can see that $h(r)$ is always decreasing by using the chain rule on $\mu_P(h,r) - \mu_E(h,r) = 0$, just as in the previous argument.

The growth maximizing strategy must occur somewhere on the curves described by $(g(r), r)$ and $(h(r), r)$. The functions $\mu_1(r) = \mu(g(r), r)$ and $\mu_2(r) = \mu(h(r), r)$ are
continuously differentiable functions of $r$ except where $g(r) = h(r)$ (which must exist by the intermediate value theorem). Therefore, the only place where $\mu$ can have a maximum is at the place where $g(r)$ and $h(r)$ intersect. This is the strategy that leads to equality of all the growth rates. We refer to this strategy, as a function of environmental conditions, as $\big(r_m(P, I, T), E_m(P, I, T), L_m(P, I, T)\big)$. Using this strategy, we can predict the stoichiometry of
the functional components of the phytoplankton population in a given environment.

We assume that real phytoplankton populations cluster near the optimal strategy in the local environment (Norberg et al., 2001):

$$(E_m, r_m) = \text{argmax}_{(E,r)}\mu. \tag{18}$$

For all values of environmental parameters used in this study, the unique maximum of the growth rate occurs for the set of parameter values that lead to co-limitation by nutrients,
photosynthesis, and biosynthesis, analogously to the predictions of Klausmeier and co-workers (2004). The optimal strategy determines the model prediction of the C:P of functional components in a given environment by taking the quotient of the carbon and phosphorus quotas.

**Table 2.** *Physiological Model Constants.*

| PARAMETER | DESCRIPTION | VALUE | UNITS | SOURCE |
|---|---|---|---|---|
| $\alpha$ | Proportionality coefficient for radius | 0.12 | - | (Toseland et al., 2013) |
| $\gamma$ | Percent dry mass devoted to structure other than membrane | 0.2 | - | (Toseland et al., 2013) |
| $k_{E0}$ | Synthesis rate of biosynthesis | 0.168 | $hr^{-1}$ | (Shuter, 1979) |

| | | | | |
|---|---|---|---|---|
| | apparatus at $T_0=25$ | | | |
| $Q_{10,E}$ | $Q_{10}$ of biosynthetic apparatus | 2.0 | | (Shuter, 1979) |
| $\Phi_{M0}$ | Specific carbon cost of maintenance at $T_0=25$ | $10^{-3}$ | $hr^{-1}$ | (Shuter, 1979) |
| $Q_{10,M}$ | $Q_{10}$ of maintenance | 2.0 | - | (Shuter, 1979) |
| $Q_{10,P}$ | $Q_{10}$ of photosynthesis | 1.0 | | (Shuter, 1979) |
| $\Phi_S$ | Carbon cost of synthesis | 0.67 | - | (Shuter, 1979) |
| aP | Allometric scaling constant for VMP | $1.04 \times 10^{-16}$ | $(mol\ P)(hr)^{-1}$ | (Edwards et al., 2012) |
| bP | Allometric scaling exponent for VMP | 3.0 | - | (Edwards et al., 2012) |
| aK | Allometric scaling constant for KP | $6.4 \times 10^{-8}$ | $(mol\ P)(L)^{-1}$ | (Edwards et al., 2012) |
| bK | Allometric scaling exponent for KP | 1.23 | - | (Edwards et al., 2012) |
| $\rho$cell | Cell Density | $10^6$ | $g/m^3$ | (Shuter, 1979) |
| pdry | Fraction of dry mass in cell | 0.47 | - | (Toseland et al., 2013) |
| $\alpha E$ | Fraction of dry mass in biosynthetic apparatus devoted to ribosomes | 0.55 | - | (Toseland et al., 2013) |
| Prib | Fraction of ribosomal mass in phosphorus | 0.047 | - | (Sterner and Elser, 2002) |
| pDNA | Fraction of cell dry mass in DNA | 0.01 | - | (Toseland et al., 2013) |
| PDNA | Fraction of DNA mass | 0.095 | - | (Sterner and Elser, 2002) |

| | | | | |
|---|---|---|---|---|
| | in phosphorus | | | |
| k1 | Specific Efficiency of Carbon Fixation Apparatus | 0.373 | $hr^{-1}$ | (Talmy et al., 2013) |
| k2 | Specific Efficiency of Electron Transport Apparatus | 0.857 | $hr^{-1}$ | (Talmy et al., 2013) |
| αPh | Light Absorption | 1.97 | $m^2/gC$ | (Morel and Bricaud, 1981) |
| φM | Maximum Quantum Efficiency | $10^{-6}$ | gC/μmol photons | (Falkowski and Raven, 1997) |
| $m_{lip}$ | Fraction of cell membrane composed of lipids | 0.3 | - | (Toseland et al., 2013) |
| $m_{prot}$ | Fraction of cell membrane composed of protein | 0.7 | - | (Toseland et al., 2013) |
| $p_{lip}$ | Fraction of cell dry mass in storage lipids | 0.1 | - | (Sterner and Elser, 2002) |
| $p_{carb}$ | Fraction of cell dry mass in storage carbohydrates | 0.04 | - | (Sterner and Elser, 2002) |
| $C_{DNA}$ | Fraction of DNA mass in Carbon | 0.36 | - | (Sterner and Elser, 2002) |
| $C_{rib}$ | Fraction of ribosomal mass in Carbon | 0.42 | - | (Sterner and Elser, 2002) |
| $C_{prot}$ | Fraction of protein mass in Carbon | 0.53 | - | (Sterner and Elser, 2002) |
| $C_{lip}$ | Fraction of lipid mass in Carbon | 0.76 | - | (Sterner and Elser, 2002) |
| $C_{carb}$ | Fraction of carbohydrate mass in Carbon | 0.4 | - | (Sterner and Elser, 2002) |

The carbon quota is calculated as:

$$Q_C = \frac{\left(\frac{m_{lip}\alpha}{r}p_{lip}C_{lip} + p_{carb}C_{carb} + \alpha_E E C_{\text{rib}} + \left((1-\alpha_E)E + L + \frac{m_{prot}\alpha}{r}\right)C_{prot} + p_{\text{DNA}}C_{\text{DNA}}\right)}{\frac{4}{3}\pi r^3 \rho_{\text{cell}}p_{\text{dry}}}. \quad (19)$$

Here we see the contributions of carbon contained in both functional and storage pools, the latter of which are assumed to occupy a fixed fraction of the cell independent of the environment (but linked to cell size).

Measurements of cellular P partitioning indicate that the ribosomal RNA can sometimes contribute only 33% of the total P quota (Garcia et al., 2016). The additional phosphorus includes membrane phospholipids and storage compounds, each of which can be up- or down-regulated in response to phosphorus availability in the environment. To model this phenomenon, we assume the existence of an additional stored P pool, whose

size is a linear function of environmental P, or:

$$(P:C)_{storage} = \epsilon[P], \quad (20)$$

where $\epsilon$ is determined by the best fit to the Martiny et al. (2014) data. Our model then predicts C:P as:

$$C:P = \frac{1}{(P:C)_{(E_m,r_m)} + \epsilon[P]}. \quad (21)$$

The model parameter $\epsilon$ is calculated by minimizing the residuals of the P:C ratio predicted

by Eq.19 in comparison to the global data-set on particulate organic matter stoichiometry compiled by Martiny and others (2014). To maintain consistency with the linear regression model of Galbraith and Martiny (2015), we restrict the dataset to observations from the upper 30 meters of the water column containing particulate organic phosphorus and carbon concentrations of greater than 5 $nM$. Observations from the same station and the

same day, but at different depths in the water column are averaged together. The P:C ratio of the functional apparatus is calculated using irradiance, $T$, and [P] data from the World Ocean Atlas (Garcia et al., 2014; Locarnini et al., 2013; oceancolor.gsfc.nasa.gov/data/10.5067/AQUA/MODIS/L3B/PAR/2014/), which are used to estimate environmental conditions at the location and date of particulate organic matter

measurements. Light levels are computed by averaging irradiance over the top 50 meters of the water column, assuming an e-folding depth of 20 meters. Linear regression determines $\epsilon = 2500$ M $^{-1}$ which fits the data with an $R^2 = 0.28$. All parameters for the model are listed in Table 2.

### 460 2.3 Experimental Design

To address how changing environmental conditions affected stoichiometric ratios, carbon export, and pCO$_{2,\text{atm}}$ we performed two tests; a change in nutrients and a change in sea surface temperature. These tests allowed us to observe how the relationships between environmental conditions, carbon export and pCO$_{2,\text{atm}}$, depend on the mechanisms

responsible for stoichiometric variation in the ocean. In order to account for the effects of

particulate inorganic carbon (PIC) export, we multiply model predicted C:P$_{export}$ by 1.2, consistent with previous studies (Broecker, 1982; Sarmiento and Toggweiler, 1984).

The first set of numerical experiments examined the sensitivity of pCO$_{2,atm}$ to nutrient availability in the tropical and subtropical boxes for each of the three stoichiometric models. We varied tropical [P] from 0.15 to 1.5 μM and subtropical [P] from 1x10e$^{-3}$ to 0.5 μM by adjusting the implied biological export and determined the equilibrium pCO$_{2,atm}$ values.

The second set of experimental tests was done to quantify how temperature modifies carbon export and pCO$_{2,atm}$ for each stoichiometric model. Temperature influences carbon cycling in two ways within our model: through the solubility of inorganic carbon in seawater and through changes in phytoplankton stoichiometry within the temperature-only and multi-environmental models. Due to the well-known effects of temperature on CO$_2$ solubility, it is generally predicted that there is to be a positive feedback between pCO$_{2,atm}$ and temperature mediated by declining CO$_2$ solubility at high temperatures. To 480 study the relative strengths of the temperature solubility feedback and the temperature regulation of C:P feedback, we performed a numerical experiment in which we varied the sea surface temperature by five degrees in either direction of modern sea surface temperature. This represents a plausible range of variation under both ice-age and anthropogenic climate change scenarios. We varied tropical temperature from 21° to 31°C 485 and subtropical temperature from 19° to 29°C, determining equilibrium pCO$_{2,atm}$ values for combinations of temperature conditions.

## 3 Results

To quantify the linkages between phytoplankton physiology, elemental stoichiometry, and 490 ocean carbon cycling, we divide our results into two parts. The first is a direct study of the stoichiometric models, comparing their predictions about the relationship between stoichiometry and environmental conditions, and in the case of the trait-based model, illustrating how cellular physiology is predicted to vary across these conditions. In the second part, we show how variable stoichiometry influences carbon export and pCO$_{2,atm}$, 495 under changing phosphorus concentrations and temperature. Within these results, we distinguish the influence or lack thereof off the three distinct biomes; in particular the equatorial upwelling regions and the macronutrient depleted subtropical gyres.

### 3.1 Multi-environmental and physiological controls on plankton stoichiometry

Our multi-environmental model captured several major mechanisms hypothesized to be environmental drivers of C:P ratios including a temperature dependence of many cellular processes, a link between growth rate and ribosome abundance, and storage drawdown during nutrient limitation. The predicted relationship between environmental conditions and C:P can be understood through the environmental regulation of three factors: (i) the 505 balance between photosynthetic proteins and ribosomes, (ii) the cell radius and associated allocation to structural material, and (iii) the degree of phosphorus storage. Our model predicted that for an optimal strategy, specific protein synthesis rates will match specific rates of carbon fixation. Thus, the ratio of photosynthetic machinery to biosynthetic machinery is primarily controlled by irradiance and temperature. Increases in light levels 510 lead to higher photosynthetic efficiency, higher ribosome content, smaller cells (due to a

lower requirement for photosynthetic machinery), and lower C:P ratios (Figure 5). The response of C:P to light levels predicted by our model was muted in comparison to other subcellular compartment models because we separately modeled electron transport and carbon fixation (Talmy et al., 2013), and our predictions were consistent with the weak
relationship between irradiance and C:P (Thrane et al., 2016) (Figure 5A).

Increases in temperature increase the efficiency of biosynthesis, but not photosynthesis. Therefore elevated temperature lead to a reduced ribosome content relative to photosynthetic proteins and higher C:P ratios (Figure 6A). There leads to a non-monotonic, concave relationship between temperature and cell size, which is due to a
subtle interaction between biosynthesis efficiency (which varies greatly with temperature), maintenance costs, and size dependent uptake rates.

Nutrient concentrations do not affect the ratio of biosynthetic to photosynthetic machinery but positively relate to both P storage and cell radius. Cell radius directly influences the specific rate of nutrient uptake, and indirectly biosynthesis and
photosynthesis as the cell membrane and wall affects the space available for other investments. This effect is pronounced in oligotrophic conditions ([P] < 100nM). Here, cell radius declines below 1μm resulting in decreasing allocations to both photosynthesis and biosynthesis and elevated C:P ratios. Higher values of the cell radius are observed in nutrient rich conditions.
P concentrations also influenced C:P through the direct control of P storage. We plotted the relative contribution of the storage compartment and the functional compartment to the P quota, as a function of environmental conditions. The impact of the residual pool on the overall size of the P pool is heavily dependent on environmental conditions, varying from a minimum of close to 0% to a maximum of just under 50%. In the
vast majority of the parameter range considered here, the contribution of the residual pool is much more modest (10-20%). High values occur when phosphorus is available and the temperature is high. In these conditions, ribosomal contributions are decreased but the residual contribution is high. In cold water, high P ecosystems, the residual contribution is approximately 25%, and in oligotrophic ecosystems it is close to 0. Thus, C:P was predicted
to be a decreasing function of [P] with two distinct regimes: a moderate sensitivity regime for [P] above 100nM, and a high sensitivity regime for [P] below 100nM.

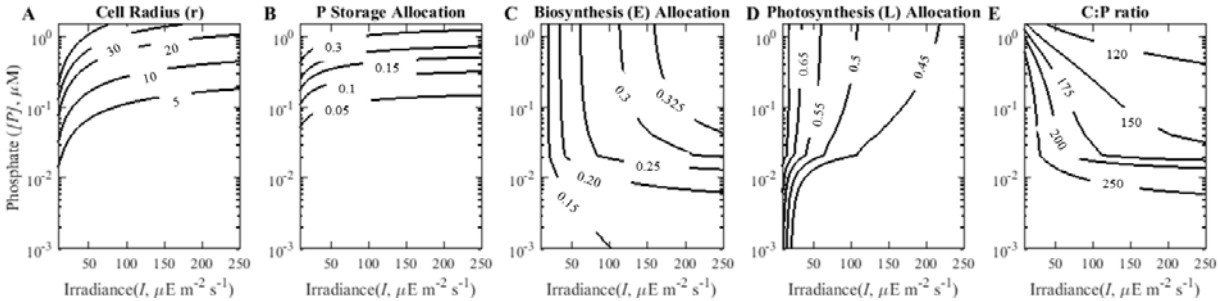

*Figure 5: Influence of phosphate concentration and irradiance on cellular stoichiometry and cellular*
*traits, at a constant T = 25 ℃. A) Cell radius (r). B) P storage allocation. C) Biosynthesis allocation. D) Photosynthesis (L) allocation. E) The C:P ratio. As irradiance increases, there is a tendency towards greater allocation to biosynthesis and lesser allocation to photosynthesis, which leads to lower C:P ratios. When phosphorus is very low, there is a large decrease in both biosynthesis and photosynthesis allocations due to the large relative allocation to the cell membrane. C:P ratios are inversely proportional to phosphorus*
*concentration, driven by an increase in luxury storage and ribosomal content as P increases.*

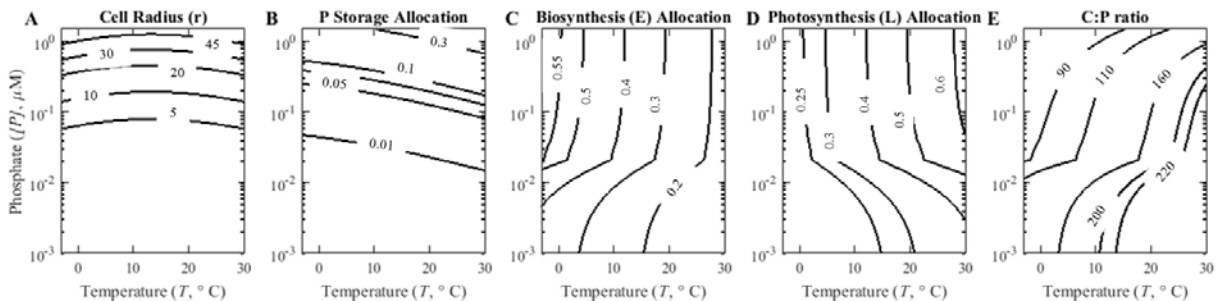

***Figure 6: Influence of phosphate concentration and temperature on cellular stoichiometry and cellular traits, at a constant irradiance I =*** $50\mu E m^{-2} s^{-1}$. *A) Cell radius (r). B) P storage allocation. C) Biosynthesis allocation. D) Photosynthesis (L) allocation. E) The C:P ratio. Consistent with the translation compensation hypothesis, increases in T led to a reduction in the allocation to biosynthesis and an increase in C:P.*

We next used the outcome of the trait model as a multi-environmental model to predict C:P ratios in the modern ocean based on annual mean light, *T*, and [P]. Our predictions reproduced the global pattern (Martiny et al., 2013b) with C:P ratios above the Redfield ratio in subtropical gyres and C:P ratios below the Redfield ratio in equatorial and coastal upwelling regions and subpolar gyres (Figure 7A). Additionally, our model also reproduced basin-scale stoichiometric gradients among similar biomes in each ocean, predicting the highest C:P ratios in the western Mediterranean Sea and the western North Atlantic Subtropical Gyre, and somewhat elevated C:P ratios in the South Atlantic Subtropical Gyre as well as the North and South Pacific Subtropical Gyres.

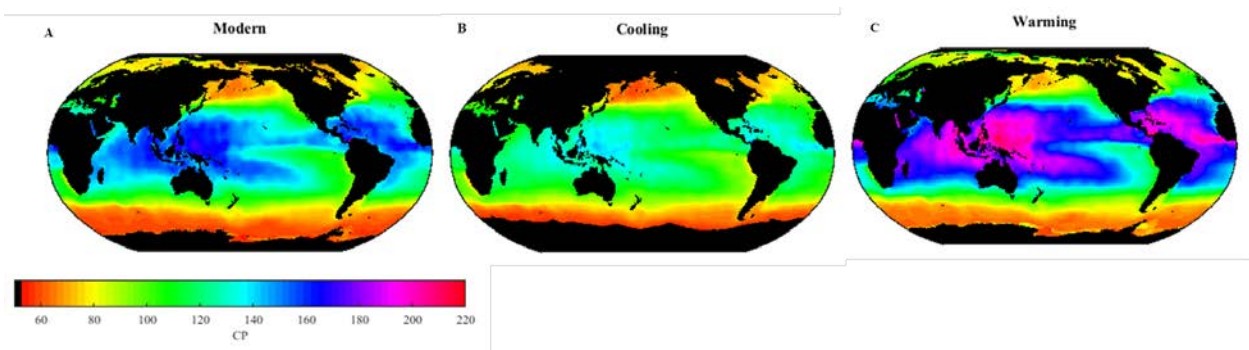

***Figure 7: Predicted C:P ratios in the global ocean in differing climatic regimes.*** *A) C:P ratio under modern ocean conditions. Large differences in C:P are predicted between distinct types of ocean biome, with low C:P in equatorial upwelling regions and subpolar gyres, and high C:P in subtropical gyres. Regional differences between biomes of similar type are observed as well, with the low phosphorus Atlantic having a higher C:P than the Pacific. B) C:P ratio under cooling temperature conditions (-5°C from the modern ocean). C) C:P ratio under warming temperature conditions (+5°C from the modern ocean). Each 5 degree change leads to a shift of 15% in the mean C:P ratio of organic matter.*

To study the potential impact of sea surface temperature on phytoplankton resource allocation and stoichiometry, we used our multi-environmental model to predict C:P in ocean conditions both five degrees colder (Cooling environments) and warmer (Warming environments) than the modern ocean. According to our model, a five-degree increase (or decrease) in sea surface temperature would cause a 15% rise (or fall) in C:P ratios (Figure 7). This sensitivity suggested that the relative effect of *T* on biochemical processes could

have large implications for biogeochemical cycles, making it important to determine the relative importance of physiological mechanisms regulating C:P ratios.

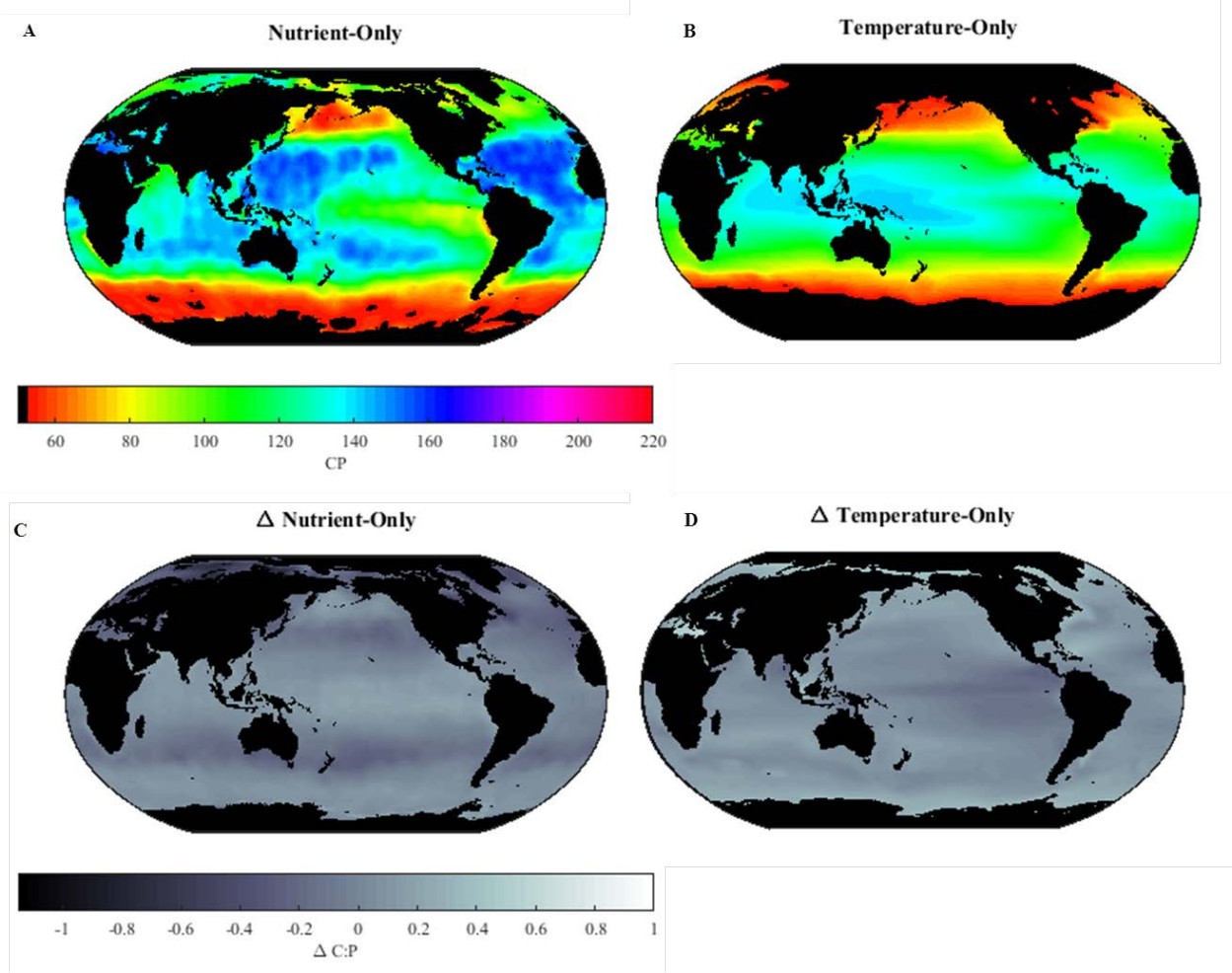

***Figure 8: Comparison of C:P between the multi-environmental model and the nutrient-only model and temperature-only model.*** *The upper panels show predicted C:P for the global ocean under the nutrient-only (A)*
*and temperature-only (B) models, and the lower panels show the normalized difference, i. e.* $\frac{C:P_{subcell} - C:P_{other}}{C:P_{subcell}}$,

*between the C:P in the subcellular model (C, D).*

We compared the multi-environmental model to the predictions made by two other models: the nutrient-only model used by the Galbraith and Martiny model (2015), and our
temperature-only model modified from Yvon-Durocher and co-workers (2015). These two models also successfully predicted the qualitative pattern of stoichiometric variation in the ocean, but were unable to replicate the full range of variation observed in the data (Figure 8). In particular, there were mismatches in the North Atlantic Subtropical Gyre and the Southern Ocean, where the C:P ratio is at the extreme (Figure 8A, B). The nutrient-only
model had a tendency to predict lower C:P ratios than the multi-environmental model in warm tropical and subtropical waters, and predict higher C:P ratios in cold waters (Figure 8A). This difference is driven by the *T* sensitivity of biosynthesis in the multi-

environmental model, leading to increasing C:P in all warm water regions and decreasing C:P in cold water regions (Figure 8C ). The multi-environmental model predicted a wider
range of C:P in the ocean. The temperature-only model overall had higher C:P ratios globally compared with the multi-environmental model (Figure 8B) but suggested lower C:P in the gyres and higher C:P in high latitudes, especially in the Southern Ocean (Figure 8D).

## 610 **3.2 Impact of nutrient availability on carbon export and atmospheric $pCO_2$**

We next quantified the impact of nutrient availability in the tropics and subtropics on stoichiometry, carbon export, and $pCO_{2,atm}$ (Figure 9A-L). Using a constant Redfield model (or the temperature-only model), we replicated the previously observed approximately linear relationship between surface [P] and $pCO_{2,atm}$ (equivalent to how pre-formed [P] will
influence $pCO_{2,atm}$) (Ito and Follows, 2003; Sigman and Boyle, 2000). We found that [P] drawdown in the subtropical box had a greater impact on carbon export, since export from the high-latitude box was not enhanced by the [P] supply from the subtropical box (Figure 9A, D, G). In the Redfield model, $pCO_{2,atm}$ appeared to be much more sensitive to subtropical [P] than tropical [P], which was partially due to enhanced carbon export in the
subtropical box and partially due to the larger surface area of the subtropical box (implying a greater potential for $CO_2$ exchange) (Figure 9J).

In contrast to the predictions made using Redfield stoichiometry, when we used the nutrient-only model for phytoplankton stoichiometry, we observed a non-linear relationship between surface [P] and $pCO_{2,atm}$ (Figure 9B, E, H, K). At fixed tropical [P],
there was a strong relationship between subtropical [P] drawdown, export, and $pCO_{2,atm}$ in accordance with the findings of Galbraith and Martiny (2015) (Figure 9B, E,H). The total decline in $pCO_{2,atm}$ as subtropical [P] declined from 0.4 μM to $1x10^{-3}$ μM could be more than 60 ppm, which was more than twice the decline that occurred in the fixed stoichiometry experiment (Figure 9K). We found a non-linear monotonic relationship
between tropical [P] and $pCO_{2,atm}$: when tropical [P] was high, declines in tropical [P] led to lower carbon export and increased $pCO_{2,atm}$. However, this trend reversed when tropical [P] was further drawn down (Figure 9K). The counter intuitive decline in $pCO_{2,atm}$ with higher export from tropics was driven by a teleconnection in nutrient delivery between the subtropical and tropical boxes. Increases in export in the tropical box due to [P] drawdown
decreased the supply of [P] to the subtropics, which led to a decrease in the more efficient (higher C:P) subtropical export. Thus, the nutrient-only model predicted a greater decrease in subtropical export than the increase in tropical export.

The multi-environmental model also predicted a non-linear relationship between P draw down, carbon export, and $pCO_{2,atm}$. However, the pattern was somewhat distinct from
that of the nutrient-only model results (Figure 9C, F, I, L). First, subtropical [P] drawdown had a nonlinear relationship with $pCO_{2,atm}$: when subtropical [P] was high, declines in tropical [P] led to slight declines in $pCO_{2,atm}$, and when subtropical [P] is low, small declines in tropical [P] lead to large declines in $pCO_{2,atm}$. This intensification of the relationship between subtropical [P] and $pCO_{2,atm}$ was due to the nonlinear relationship between [P]
and C:P predicted by the trait-based model (Figure 9I). The multi-environmental model predicted extremely high tropical export, but only when [P] was lower than 0.05 μM (Figure 9C, F, I). Second, the effect of tropical [P] levels on $pCO_{2,atm}$ was strongly modulated by subtropical [P], reversing from a negative to a positive relationship as subtropical [P]

declines (Figure 9I, L). The difference between the nutrient-only model and the multi-environmental model arose because the multi-environmental model incorporated a temperature impact on resource allocation and elemental ratios. Although we were not varying temperature in these experiments, we did represent regional temperatures differences between the different boxes. The result is that a large stoichiometric contrast between the tropical and sub-tropical regions only arose when there was a large difference in nutrient levels between the two regions (Fig. 9L). However, both the nutrient-only model and the multi-environmental model predicted that carbon export and $pCO_{2,atm}$ were sensitive to the interaction between regional nutrient availability and $C:P_{export}$.

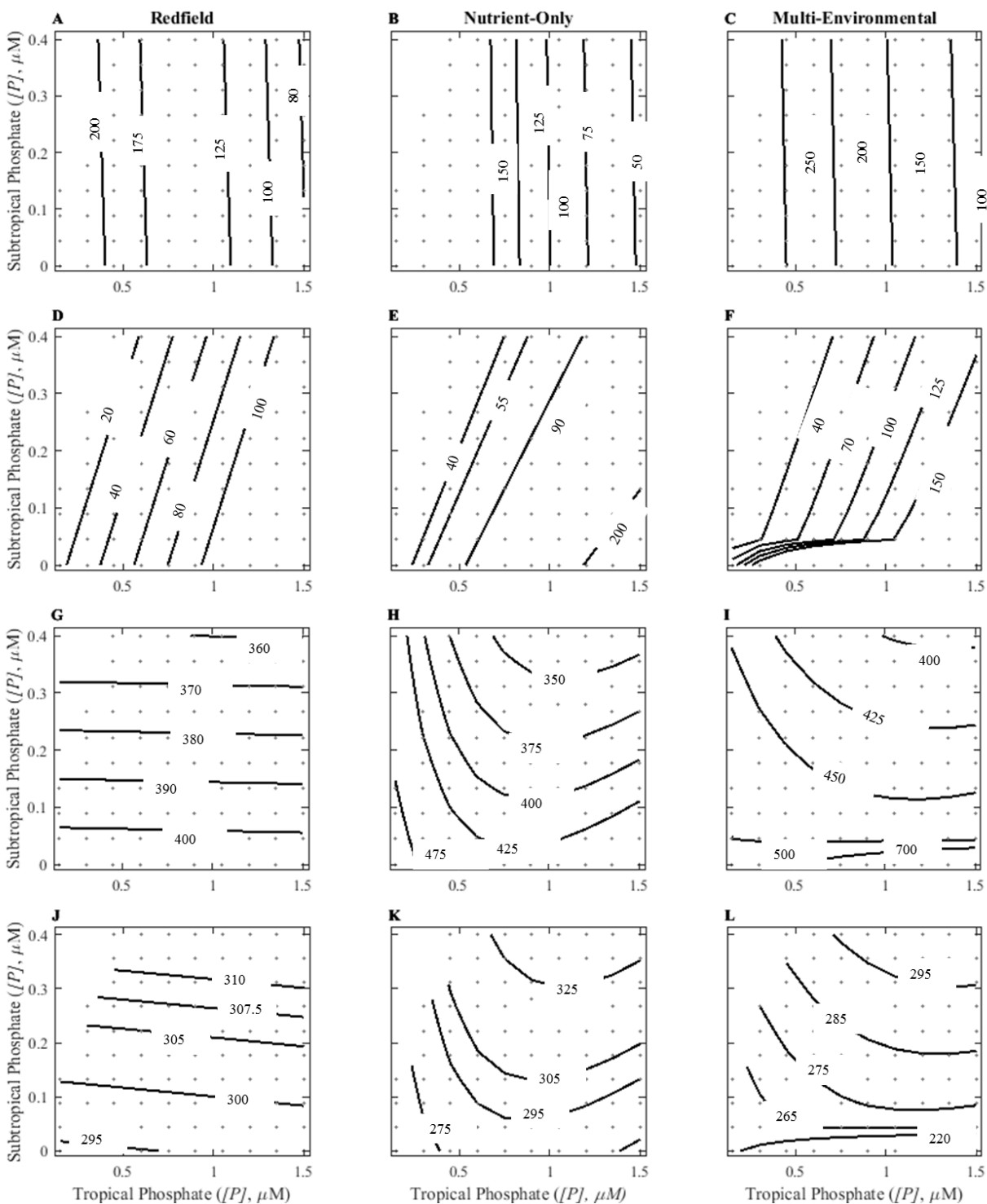

*Figure 9: Carbon export (Tmol C yr⁻¹) and pCO₂,atm (ppm) in changing surface phosphate concentrations.*
*Columns correspond to type of stoichiometry; Redfield (Left), nutrient-only (Middle), and multi-environmental model (Right). Rows correspond to either tropical carbon export (A through C), subtropical carbon export (D through F), total carbon export (G through I) or atmospheric pCO₂ (J through L). The grey points represent where pCO₂,atm was calculated, between spaces are interpolated.*

### 3.3 Interactive effect of temperature on stoichiometry, carbon export and atmospheric pCO$_2$

We next quantified the impact of sea surface temperature (SST) in the tropics and subtropics on C:P$_{export}$, carbon export, and pCO$_{2,atm}$ (Figure 10A-D). The Redfield model
predicts that increases in temperature lead to a decline in the solubility of CO$_2$ in seawater and consequently an increase in pCO$_{2,atm}$ from 288 to 300 ppm ($\Delta$ pCO$_{2,atm}$ = 12) (Figure 10A). This feedback was present with the same strength in the nutrient-only model (with no $T$ dependence on C:P), in which pCO$_{2,atm}$ ranged from 268 to 280 ppm ($\Delta$ pCO$_{2,atm}$ = 12) (Figure 10B).

In contrast to the Redfield and nutrient-only models, the temperature-only model predicted a negative linear relationship between pCO$_{2,atm}$ and tropical sea surface $T$ and a positive linear relationship between pCO$_{2,atm}$ and subtropical sea surface $T$ (Figure 10C). The decline in pCO$_{2,atm}$ with tropical SST was driven by an enhancement of export due to increased C:P at higher temperatures (Figure 11). At 5°C below modern ocean temperature,
the model predicted C:P in the tropics was 131 and subtropical was 121, resulting in a pCO$_{2,atm}$ of 305 ppm. At 5°C above modern ocean temperature, the model predicts a C:P ratio in the tropics of 189 and C:P ratio of 175 in the subtropics, resulting in a pCO$_{2,atm}$ of 263 ppm. Tropical SST had more impact with $\Delta$ pCO$_{2,atm}$ = 41 ppm compared to subtropical SST with a $\Delta$pCO$_{2,atm}$ ranging from 4 to 5 ppm (Figure 11).

Similar to the temperature-only model, the multi-environmental model predicted a negative linear relationship between pCO$_{2,atm}$ and tropical SST and a positive linear relationship between pCO$_{2,atm}$ and subtropical SST (Figure 10D). The decline in pCO$_{2,atm}$ with tropical SST was driven by an enhancement of export due to increased C:P at higher $T$s (Figure 11). In the subtropical region, the expected increase in export was mitigated by a
decline in solubility. At 5°C below modern ocean temperature, the trait-based model predicted that C:P in the tropics was 147 and that C:P in the subtropics was 155, resulting in an increase of pCO$_{2,atm}$ to 279 ppm (Figure 11). Variation in tropical SST over a 10°C span led to a significant decline in pCO$_{2,atm}$, with a $\Delta$ pCO$_{2,atm}$ of approximately 46, and tropical C:P ranging from 147 to 210 (Figure 11). Because the subtropical box has a large surface
area, the decrease in surface CO$_2$ solubility at high temperatures is sufficient to overcome the increase in export due to higher C:P leading to a positive relationship between pCO$_{2,atm}$ and subtropical temperatures.

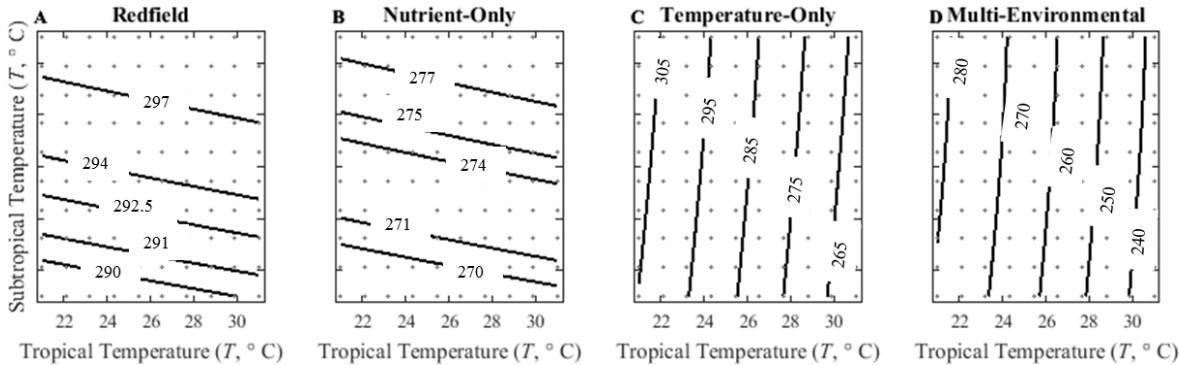

*Figure 10: pCO$_{2,atm}$ (ppm) as a function of changing surface temperature concentrations. Based on A) Redfield (fixed) stoichiometry model, B) nutrient-only stoichiometry model, C) temperature-only stoichiometry model, and D) multi-environmental stoichiometry model.*

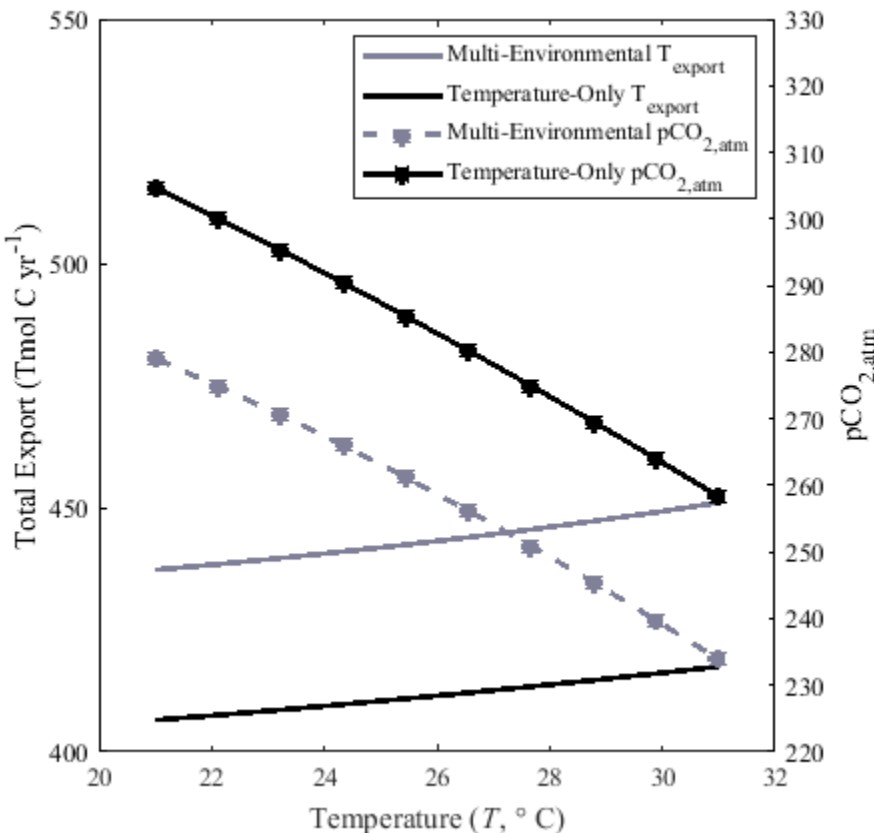

*Figure 11: The effect of changing sea surface temperature (°C) on pCO$_{2,atm}$ and total carbon export (Tmol* 705 *C yr$^{-1}$) in the temperature-only and multi-environmental model. Phosphate concentrations are 0.3 μM in the tropical and 0.05 μM in the subtropical box. Multi-environmental model total carbon export is the solid gray line, and pCO$_{2,atm}$ is the dashed gray line. Temperature-only model total carbon export is the solid black line, and pCO$_{2,atm}$ is the dashed black line.*

## 710 4 Discussion

Here, we found that variable stoichiometry of exported organic material moderates the interaction between low-latitude nutrient fluxes and ocean carbon cycling. A full connecting circulation allows for complete movement of nutrients between ocean regions resulting in strong linkages between nutrient supply ratios and cellular stoichiometric 715 ratios (Deutsch and Weber, 2012). It has been shown that the inclusion of an oceanic circulation connecting high and low-latitude regions results in a feedback effect between high-latitude nutrient export and relative nutrient supply in low-latitudes (Sarmiento et al., 2004; Weber and Deutsch, 2010). Together, the inclusion of lateral transport between ocean regions and of deviations from Redfield stoichiometry within our model led us to 720 predict the existence of strong teleconnections between the tropics and the macronutrient limited subtropics. The degree of nutrient drawdown in the tropics had a strongly non-monotonic relationship with pCO$_{2,atm}$ because this drawdown influenced both nutrient supply to the subtropics and tropical C:P. The idea of biogeochemical teleconnections has

been proposed before, but we found that variations in stoichiometry greatly enhance the importance and strength of such linkages (Sarmiento and Toggweiler, 1984). Thus biome-scale variations in phytoplankton elemental stoichiometry may change the sensitivity of the carbon pump to other phenomena that regulate patterns of nutrient drawdown. We also see that the degree of nutrient drawdown had a strong impact on predicted (and observed) C:P leading to highly non-linear controls on $pCO_{2,atm}$, whereby increased export in the tropics counter intuitively leads to increasing $pCO_{2,atm}$. Large-scale gradients in stoichiometry can alter the regional efficiency of the biological pump: [P] supplied to high C:P regions leads to a larger export of carbon than [P] supplied to low C:P regions. This lends an important role to details of ocean circulation and other processes that alter nutrient supply and phytoplankton physiological responses in different surface ocean regions. Therefore, biome-scale variations in phytoplankton elemental stoichiometry can lead to a fundamental change in the partitioning of carbon between the atmosphere and the ocean.

We have created a box model to simulate the impact of the low latitude stoichiometric ratios, its environmental controlling factors, and the relationships to $pCO_{2,atm}$. Low latitude phosphorus concentrations can be set in one of two fashions; through iron limitation and through nutrient supply. Here we will briefly discussion how iron limitation would play a significant role on phosphorus concentrations and associated C:P. The biogeochemical functioning of tropical regions are commonly influenced by iron availability in such a way that macronutrients cannot be fully drawn down by phytoplankton (Coale et al., 1996; Moore, 2004; Raven et al., 1999). The degree of nutrient drawdown has a strong impact on predicted (and observed) C:P. This environmental control on C:P could lead to highly non-linear controls on $pCO_{2,atm}$ whereby increased iron availability lead to increased [P] draw down and export in the tropics. However, as we have shown this may lead to increasing rather than commonly assumed decreasing $pCO_{2,atm}$. This link between iron and export would differ in the subtropics, where iron is thought to stimulate nitrogen levels through nitrogen fixation. This would result in elevated phosphate draw down, higher C:P and higher export. Thus, iron availability may play a complex role depending on whether there is an increased delivery in upwelling zones (leading to a potential declining global C export) or in the subtropical gyres (leading to a potential increase in global C export).

Past studies using box models have found $pCO_{2,atm}$ to be insensitive to low-latitude nutrients (Follows et al., 2002; Ito and Follows, 2003; Sarmiento and Toggweiler, 1984; Toggweiler, 1999). This phenomena was explored by DeVries and Primeau (2009), who showed that the strength of the thermohaline circulation is the strongest control on $pCO_{2,atm}$, and that changes in low-latitude export has a minor impact. Unlike our study, such earlier work relied on a uniform Redfield stoichiometry. However, we find that when stoichiometric variation is included, carbon export and $pCO_{2,atm}$ become dependent on details of low-latitude processes.

It is important to recognize that a five-box model is an incomplete description of ocean circulation, and is here used to illustrate important mechanisms, not to make precise quantitative predictions. In order for our model to adequately reflect important features of the carbon and phosphorus nutrient distributions, we had to carefully select the values of the thermohaline and wind-driven upper ocean circulations that lead to reasonable nutrient fluxes and standing stocks. The value of thermocline circulation, Tc, has been

calibrated in different box models to range from 12 to 30 Sv (DeVries and Primeau, 2009; Galbraith and Martiny, 2015; Sarmiento and Toggweiler, 1984; Toggweiler, 1999). Variations in the thermohaline circulation influence the abundance of nutrients in different boxes. Depending on the strength of this circulation, our model accumulated nutrients in the thermocline box and we tuned this parameter to most accurately mimic nutrient

variation across ocean regions. Other caveats relates to our choice of the wind driven overturning circulation and the two-way flux values. Similar to the circulation values, a wide range of two-way flux values have been used in the literature. We therefore performed sensitivity experiments to find the best value for our full model set-up but the qualitative trends observed are insensitive to the choice of such fluxes.

Nutrient availability and temperature have been alternatively proposed as drivers of variation in stoichiometric ratios in the global ocean, and the strong statistical correlation between temperature and nutrients throughout the ocean has prevented identification of the relative importance of each factor (Martiny et al., 2013b; Moreno and Martiny, 2018). We see that although temperature regulation of C:P$_{export}$ can influence

pCO$_{2,atm}$, this regulation is strongly dependent on the detailed physiological control mechanism and also generally diverge from expectations based on the solubility pump. The decrease in surface CO$_2$ solubility at elevated temperature is sufficient to overcome the increase in export due to higher C:P leading to a positive relationship between pCO$_{2,atm}$ and subtropical temperatures.  It is important to point out that the relative importance of the

two competing effect depends critically on the physical circulation of the ocean. Predicted increases in stratification are often invoked as a mechanism that would decrease the vertical supply of nutrients, which one might think would further compensate for the effect of higher C:P. However, the strength of the biological pump in the subtropics is also influenced by lateral transport of nutrients (Letscher et al., 2015) as such we argue that it is

unclear if you should expect increasing, unchanged, or decreasing C export in low latitude regions with ocean warming and stratification.  Similarly, it is unclear how increases in stratification might affect the strength of the solubility pump. The sensitivity of pCO$_{2,atm}$ to changes in subtropical surface temperatures depends critically on the volume of the ocean ventilated from the subtropics, i.e. on the volume of the thermocline box in our model. How

this volume might change in response to a warming world is a complicated dynamical problem that is beyond the scope of the present work.

Our results do not identify whether temperature or nutrient concentrations is the most important driver of phytoplankton C:P, but do suggest that the physiological effect of temperature could be important for ocean carbon cycling. Both the temperature-only and

multi-environmental models predict that temperature increases enhance tropical export, causing substantial decreases in pCO$_{2,atm}$ with temperature. This relationship is the reverse of that predicted by the nutrient-only and Redfield models, and represents a sizable potential negative feedback on carbon cycling. The multi-environmental model also predicted that C:P responds in a nonlinear fashion to [P], with significantly increased

sensitivity in highly oligotrophic conditions. Thus, a deeper understanding of the physiological mechanisms regulating phytoplankton C:P ratios are key to understanding the carbon cycle.

Our derivation of the multi-environmental model relies on several important assumptions. The growth rate in the multi-environmental model is determined by a set of

environmental conditions and quantified by the specific rate of protein synthesis, carbon

fixation, and phosphorus uptake. The effect of growth rate on stoichiometry will likely be dependent on whether light, a specific nutrient, or temperature controls growth (Moreno and Martiny, 2018). The value of specific values of $Q_{10}$ leads to uncertainty in our multi-environmental model because of the range of possible values is highly dependent on the

cell or organism being tested. In a study examining $Q_{10}$ of various processes within the cell, it was found that the $Q_{10}$ of photochemical processes ranged from 1.0 to 2.08, and for carboxylase activity of RuBisCO to be 2.66 (Raven and Geider, 1988). In addition to the high uncertainty between $Q_{10}$ values, there is high ambiguity associated with cellular inorganic P stores (e.g., polyphosphates and phospholipids) (Kornberg et al., 1999). P storage, such as

polyphosphates, can serve as both energy and nutrient storage that may be regulated by unique environmental factors.  Thus, we recognize multiple caveats within the trait-based model but expect that it improves our ability to link environmental and phytoplankton stoichiometry variation.

## 5 Conclusions

We find that processes that affect nutrient supply in oligotrophic gyres, such as the strength of the thermohaline circulation, are particularly important in setting $pCO_{2,atm}$ but via a complex link with C:P$_{export}$. By explicitly modeling the shallow overturning circulation, we showed that increased export in the tropics, which might be influenced by increased atmospheric iron dust deposition, may lead to increases, rather than decreases, in $pCO_{2,atm}$.

Increased [P] drawdown in the tropics shifts export away from the subtropical gyres, and changes the mean export C:P in the low-latitude ocean. Additionally, we find that it is difficult to separate nutrient supply and temperature controls on marine phytoplankton stoichiometry, carbon export, and $pCO_{2,atm}$ and we need better physiological experiments and field data to fully understand the relative impact of the two factors. Nevertheless, it is

likely that both play a key role in regulating phytoplankton stoichiometry, C:P$_{export}$, and ultimately ocean carbon cycling.

**Author Contribution:** ARM - creation and analysis of the box model and primary writer of manuscript. GIH - creation and development of the trait-based model, and writing. FWP -

assistance on the box model and editing of manuscript. SAL - assistance on the trait-based model and editing of manuscript. ACM - assistance on both models and writing of manuscript.

**Competing Interests:** The authors declare that they have no conflict of interest.

**Acknowledgement**
We thank Alyse Larkin at UCI for many helpful comments. This work was supported by NSF to ARM (Graduate Research Fellowship Program), FWP (OCE 1756906), SAL (OCE-1046001 and GEO-1211972), and ACM (OCE-1046297 and OCE-1559002). FWP was

supported by DOE Office of Biological and Environmental Research award DE-SC0012550. SAL was supported by the Simons Foundation Grant 395890.

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
