# Peer review of "Marine Phytoplankton Stoichiometry Mediates Nonlinear Interactions Between Nutrient Supply, Temperature, and Atmospheric CO2"

_Biogeosciences, 2017_

## Referee Comment (RC1) · Anonymous Referee #1 · 7 Nov 2017

Variability in marine phytoplankton stoichiometry can lead to differences in carbon export to the deep ocean. This manuscript expands on previous models (mainly Galbraith and Martiny, 2015 and Yvon-Durocher et al., 2015) to further our understanding of how various environmental factors lead to changes in phytoplankton stoichiometry. The authors show how incorporating factors, such as temperature, light, and phosphorus concentrations, are able to model variations across the global ocean that would otherwise be lost in more mainstream models using fixed C:P ratios. The manuscript overall nicely lays out the difference modeling approaches taken and how they ultimately change the effect on carbon export in the global ocean. I want to point out that I am not a modeler and therefore cannot asses the math presented to its full extend, but I am a

biogeochemist and can provide a critique on the science presented in this manuscript. I believe this manuscript should be published with the revisions I have outlined below.

My first issue with the manuscript in general is the lack of nitrogen. The ocean overall is nitrogen limited and it is a bit worrisome that it is never mentioned. Why was nitrogen left out? There must be a reason, and including a sentence or two to explain why it has been left out would be sufficient without having to incorporate it into the models.

My second issue is the inclusion of iron and iron deposition. It is mentioned several times throughout the manuscript and honestly seems to be thrown in haphazardly. Even calling the one region the iron-limited upwelling zone does not really make sense. I do not disagree that these regions are distinct from the subtropical gyres, but there needs to be another way to separate them. The simplest thing to do would be to remove all talk of iron and iron deposition as it does not add anything to the manuscript. If you choose though to leave it in, there needs to be more discussion and also a few references as there are currently none. I have listed below each mention of iron and have provided some input should you choose to include it.

My third issue is how phospholipids have been defined and treated within the model. The decision to functionally treat phospholipids with the storage pool needs to be justified or expanded upon as it is currently not clear. As the authors state, phospholipids are localized within the cellular membrane (defined in the model as a functional pool) and not as energy storage molecules as suggested in the text. Lipids associated with energy storage, localized within intracellular lipid droplets, are generally non-phosphorus, highly reduced, and non-polar (see Levitan et al 2014 "Remodeling of intermediate metabolism in the diatom Phaeodactylum tricornutum under nitrogen stress"). This raises the question of how the authors have defined the storage pool, is it defined as utilized by organisms for energy storage or is it only in the sense that "this is a pool where some phosphorus is stored within the cell?" L34: First mention of iron-limited tropical upwelling region. Again, I would honestly remove "iron-limited" and just call the region the tropical upwelling region. These regions become macronutrient

limited as well once you leave the immediate upwelling zone so its deceiving to just focus on iron. It is also known that the Southern Ocean is iron limited, so, again, it is deceiving to focus on iron in the upwelling regions when there are other regions that are iron limited.

L46: add "ppm" after $\sim$46

L70: add "et al." after Durocher

L76-85: Remove iron-stressed and iron-limited. The sentence "Iron deposition in the tropical upwelling. . ." is not correct. There is actually very little iron deposition to the tropics, the North African dust plume deposits iron to the tropical Atlantic but that is the one example (see Jickells et al 2005 "Global Iron Connections Between Desert Dust, Ocean Biogeochemistry, and Climate"). Iron is upwelled along the coasts in these areas along with macronutrients, but it is incorrect to call that iron deposition.

L111-115: Questions 1 and 2 seem redundant, please remove question 1 or give more detail if it is in fact different from question 2.

L135: Where in the water column are you taking the phosphorus concentration?

L207: add "et al." after Daines

L212: add "et al." after Daines

L213: Expanding on the phospholipid justification, can you explain more why you choose a zero contribution for phospholipids? Although non-P substitutes can reduce the phosphorus incorporated into P-lipids, observations suggest non-zero quantities remain. For example, 1.3 +/- 0.6% P uptake in the P-limited Sargasso are incorporated into phospholipids (Van Mooy et al 2009 – mentioned in next correction) and phospholipids make up approximately 5% of particulate organic P in the P-limited eastern Mediterranean (Popendorf et al 2011 "Gradients in intact polar diacylglycerolipids across the Mediterranean Sea are related to phosphate availability"). Might it be more appropriate to have two distinct P-lipid/total cellular P values for high and low phosphorus regions?

L215-216: For phospholipid substitution, a more appropriate reference would be Van Mooy et al 2009 "Phytoplankton in the ocean use non-phosphorus lipids in response to phosphorus scarcity" instead of Van Mooy et al 2006.

L228: add "et al." after Daines

L282: . . .that underlies the subtropical gyres and equatorial upwelling regions (labeled M), and deep waters. . .

L314: "Iron limitation is implicitly simulated through its control on the tropical [P]. . ." – how does iron control phosphorus concentrations? This is not clear in the manuscript and I personally have not come across any such research stating such. Again, if you are going to keep iron in the manuscript please provide references of where you have gotten the information and expand on the explanation of how you can make this justification.

Table 2: Please switch the columns so that Range of fhd (sv) is first and references is second (will be consistent with Table 1).

L350: "This set of experimental runs was intended to capture the effects of changing levels of iron deposition. . ." – Again, talking about iron deposition in these tropical upwelling regions does not make sense and as you have not provided references I would just remove it all together. This experiment wanted to test the sensitivity of pCO2 to nutrient availability, I believe that is a good enough reason and there is no need to mention iron limitation.

L377: Change variables to variable

L379: Remove "iron stressed"

Figures 6 and 10: I really like these figures and think you could include more in the discussion about the implications of how global temperatures will affect export. It is

a nice way to tie your work with large scale impacts on biogeochemical cycles and reiterate the importance of the study.

L477: add "the" before data

L589: remove iron-limited

L596: remove "iron deposition or"

L600: remove sentence "This observation suggests that pCO2 may have a complex link...". You honestly have not shown anything to do with iron delivery and its link to pCO2, there is nothing included in the model that I saw and again have provided zero references about iron deposition

L650: remove "thus"

L674: remove "which might be influenced by increased atmospheric iron deposition,"

L680: change separating to separate

References: There are a few references that are not mentioned in the manuscript. A couple are about iron cycling and I am curious if and where they were originally included and also possibly had more of an explanation associated with them of why you link iron to phosphorus? Cunningham and John 2017 Moore 2004 Raven and Falkowski 1999 Also, please move Van Bogelen and Neidhardt 1990 and Van Mooy et al 2008 references to after the Toseland et al 2013 reference

---

## Referee Comment (RC2) · Anonymous Referee #2 · 1 Dec 2017

Using a "classical" ocean carbon cycle box model and parameterizations of flexible elemental stoichiometry in surface ocean particulate, the authors examine the role of elemental composition in mediating the response of atmospheric CO2 to ocean temperature change. There is a significantly different sensitivirty when the particulate C:P ratio is represented by a "multi-environmental" (Ecological Stoichiometry and temperature dependent) framework compared to a fixed, Redfieldian particulate composition. Most notably, compensation between temperature sensitivity of solubility and biological pumps reduces the sensitivity to subtropical temperature change as well as reversing and enhancing the response to tropical perturbations. Variable elemental composition and phosphorus storage modify the sensitivity of atmsopheric pCO2 to the efficiency

of phosphate utilization in subtropics and tropics. The key message for is that variable elemental ratios have non-negligible impacts on the ocean's control of atmospheric CO2 and that the temperature sensitivity of solubility and stoichiometrically-mediated biological pumps have some interesting regional dependences.

I enjoyed reading this paper. I found it stimulating and thought provoking. There is alot going on. The authors combine classical carbon cycle box model with several parameterizations of elemental stoichiometry of the sinking particulate (and/or primary producers). The authors connect cellular scale physiology and global carbon cycle and have used and developed an appropriate framework with which to do so.

My criticism of the paper is that the multi-environmental model is presented very much at face value. The assumptions and construction seem very logical but the choice and constraint of the parameters is by and large opaque. In particular the relationship between the storage component of the multi-environmental model and the Galbraith and Martiny parameterization seems interesting and important but is not really discussed. How important is the storage term in controlling the overall response of the multi-environmental model? It is not at all clear from the manuscript. I feel that some clarification and discussion along these lines is important for the reader.

I found the manuscript very interesting and thought provoking. I had a number of questions, comments and need clarification on certain points which I will detail here. Some are more important than others. While my recommendation is major revision, it is clarification that I would like to see, not changes to what has been done.

1. The Sarmiento and Toggweiler and contemporary carbon cycle models focused alot on the sensitivity of atmospheric pCO2 to "high latitude" changes. This isnt discussed here - perhaps there wasnt any as configured? Some comment would be useful and interesting in this regard.

2. The stoichiometry of sinking particulate and of primary producers is certainly connected but not necessarily the same. The multi-environmental model is founded on

primary producer physiology. Perhaps this potential difference should be flagged?

3. I would have been interested to see a Droop-style model in the mix as its a relatively common tool - but there is more than enough going on here anyway.

4. I very much like the spirit of the multi-environmental model. (Though I find the name a little odd). It accounts for the role of cell size in mediating nutrient affinity and cell composition (contribution of cell wall material). It was not made clear how sensitive the final parameterization or the outcomes of the box model are to the assumed cell size. Nor could I find any information about the cell size assumed (or modeled?) in the simulations.

5. The photosynthesis parameterization and allocation scheme is very reminiscent of Geider's models in spirit and mathematical form. What is the relationship?

6. The statement at line 235 that the "unique maximum of the growth rate occurs for the set of parameters that lead to co-limitation by nutrients, photosynthesis and biosynthesis" is very interesting and intriguing. Is that an emergent property? Is it obvious that it should be this way? I would have liked to hear more about this.

7. Phosphorus storage seems to be very important. Equation (13) controls a residual storage pool that constrains the parameterized stoichiometry to match the observed relationship between phosphate and particulate stoichiometry, as I understand it. Thus it strongly mirrors the Galbraith and Martiny model of equation (1). For me, some key questions concern this aspect of the model: How significant in the overall control of model stoichiometry is this component? If it dominates, then I could view the multi-environmental model in some way as a combination of the Galbraith and Martiny model with a temperature sensitivity. Or does the more mechanistic and detailed physiology have a significant role? Either way, I think the mechanistic model is valuable and interesting but I would like to understand how much the results are driven by the storage of phosphorus. Its important for a number of reasons and I feel that this should be clearly discussed.

8. A small thing, but I had to stop and think about equation (1) because [P]o has different dimensions than [P]: the former is a ratio and the latter a concentration. I think it would be much clearer and more appropriate to denote [P]o as in (13), with a symbol in accord with other variables that are ratios.

9. The box model formulation makes sense; the inclusion of the thermocline reservoir is important for the sensitivity to changes in the subtropical surface. Some small details: how is the carbonate system solved? Is alkalinity fixed or is there an implicit carbonate pump?

10. The model doesnt resolve nitrogen, and I would expect that the allocation of nitrogen in proteins and pigments would be an important factor, perhaps more so than phosphorus. Does this actually matter? A comment on this would be helpful.

11. Why does temperature affect biosynthesis but not photosynthesis (line 398) imposed from empirical observations? The model description tells us that Q10=2 for temperature dependences, but there is no discussion with regard to phootosynthesis. Why this choice?

12. The discussion of sensitivity to cell radius in lines 400-410 doesnt tell us what is the cell radius (or distribution of) in the model? Is it imposed or modeled (I presume the former but nothing is said in the paper). This should be clear.

13. Figures 3,4,5 are a bit small and fuzzy when printed.

14. I'd really like to understand how important the storage term is in the overall control of figures 4 and 5. We see the variation in C:P and the relative allocation to biosynthesis and photosynthesis, but its not clear how important the latter is to the former.

15. What is the cell size in the box model simulations? Is it imposed? Does it vary? How sensitive are results to r?

16. The model is P based. However, as is alluded to in the manuscript, nitrogen and iron dynamics are important. Indeed P is found to be the proximal limiting resource

in only a few areas of the global ocean, with N and Fe controlling things locally. So how does this affect the relevance of the model? Wouldnt N and Fe dynamics be more important at the individual scale? Would this (does this) mean that storage is most signficant for P:C? Again, understanding the significance of storage for the outcomes here is very important.

17. Line 464: "nutirent"

18. The contrasting temperature sensitivities of tropical and subtropical perturbations is very interesting. The dominance of the solubility term in subtropical responses is ascribed to the "large surface area" of the subtropical region (line 563). I dont think thats true: I think its because the subtropical surface feeds the subtropical thermocline which reprsents a sigifnicant contribution to global water volume. Hence, changes in subtropical solubility have significant leverage. Since tropical waters dont directly feed into any subsurface water mass, they do not have the same leverage. This is why the resolution of the thermocline box is important. The classic Harvardton Bear box models did not resolve the thermocline and so found very low sensitivity to subtropical perturbations relative to 3D circulation models. Resolving the thermocline in the box models brings them into consistency (this was the point of Follows et al, 2002). I thought this was why the authors had chosen the configuration which resolves a thermocline reservoir.

---

## Author Comment (AC1) · 24 Dec 2017

We gratefully thank Referee #1 for their time, constructive comments, and suggestions to our manuscript. Below we have a detailed response to each comment posed by Referee #1. We have amended the manuscript in hopes that it will be much improved and our study presented clearer.

Anonymous Referee #1

Variability in marine phytoplankton stoichiometry can lead to differences in carbon export to the deep ocean. This manuscript expands on previous models (mainly Galbraith

and Martiny, 2015 and Yvon-Durocher et al., 2015) to further our understanding of how various environmental factors lead to changes in phytoplankton stoichiometry. The authors show how incorporating factors, such as temperature, light, and phosphorus concentrations, are able to model variations across the global ocean that would otherwise be lost in more mainstream models using fixed C:P ratios. The manuscript overall nicely lays out the difference modeling approaches taken and how they ultimately change the effect on carbon export in the global ocean. I want to point out that I am not a modeler and therefore cannot asses the math presented to its full extend, but I am a biogeochemist and can provide a critique on the science presented in this manuscript. I believe this manuscript should be published with the revisions I have outlined below.

1) My first issue with the manuscript in general is the lack of nitrogen. The ocean overall is nitrogen limited and it is a bit worrisome that it is never mentioned. Why was nitrogen left out? There must be a reason, and including a sentence or two to explain why it has been left out would be sufficient without having to incorporate it into the models.

OUR RESPONSE »> This is a very important point and the reviewer comments make it clear that the justification of using P as a representative of nutrient availability needs to be clarified in our manuscript. The underlying reason for picking P rather than N is linked to ideas outlined by Tyrrell, 1999. On long time-scales, P is commonly considered the ultimate limiting nutrient whereas N is only limiting productivity and export on short time-scales. On long time-scales, nitrogen fixation/denitrification will presumably adjust the N inventory. Our modeling is focused on long term steady-state outcomes and we would like to avoid issues associated with modeling the N cycle (like getting N-fixation and denitrification rates correct). Thus, we chose to use P as a representative for nutrient availability. However, we do recognize that the reality may be more complex and hope to add an explicit nitrogen (and Fe) cycle in the future.

We have amended the manuscript to address this concern: "Phosphorus is used to represent the role of nutrient availability in controlling stoichiometry and C export. We chose this over N to avoid having to include a parameter rich N cycle. Furthermore, P

rather than N is commonly regarded as the ultimate limiting nutrient (Tyrrell 1999) and thus P availability represents the long-term steady-state biogeochemical equilibrium."

2) My second issue is the inclusion of iron and iron deposition. It is mentioned several times throughout the manuscript and honestly seems to be thrown in haphazardly. Even calling the one region the iron-limited upwelling zone does not really make sense. I do not disagree that these regions are distinct from the subtropical gyres, but there needs to be another way to separate them. The simplest thing to do would be to remove all talk of iron and iron deposition as it does not add anything to the manuscript. If you choose though to leave it in, there needs to be more discussion and also a few references as there are currently none. I have listed below each mention of iron and have provided some input should you choose to include it.

OUR RESPONSE »>We agree with the reviewer on this point and realize that the references to Fe limitation are confusing. Thus, we have removed the labeling of iron-limited regions in the manuscript. Now, we only introduce the concept of iron limitation in the discussion as a factor contributing to setting surface macronutrient concentrations in tropical ecosystems.

3) My third issue is how phospholipids have been defined and treated within the model. The decision to functionally treat phospholipids with the storage pool needs to be justified or expanded upon as it is currently not clear. As the authors state, phospholipids are localized within the cellular membrane (defined in the model as a functional pool) and not as energy storage molecules as suggested in the text. Lipids associated with energy storage, localized within intracellular lipid droplets, are generally non-phosphorus, highly reduced, and non-polar (see Levitan et al 2014 "Remodeling of intermediate metabolism in the diatom Phaeodactylum tricornutum under nitrogen stress"). This raises the question of how the authors have defined the storage pool, is it defined as utilized by organisms for energy storage or is it only in the sense that "this is a pool where some phosphorus is stored within the cell?"

OUR RESPONSE »> We agree that this can be confusing. Due to the similarity in behavior of P-lipids and P-storage (no other types of storage molecules like lipids or carbohydrates are considered here), they were treated as the same in the model to save parameters. To address this issue, we have attempted to clarify this issue in the manuscript.

The manuscript now reads as follows: "Phytoplankton can substitute sulfoquinovos-diaglycerol (SQDG) for phospholipids in their cell membranes under low P conditions (Van Mooy et al., 2009). Similarly, P storage molecules are also regulated by P availability. Thus, we here assume that phospholipids and P-storage exhibit the same behavior and thus model-wise treated as one pool."

4) L34: First mention of iron-limited tropical upwelling region. Again, I would honestly remove "iron-limited" and just call the region the tropical upwelling region. These regions become macronutrient limited as well once you leave the immediate upwelling zone so its deceiving to just focus on iron. It is also known that the Southern Ocean is iron limited, so, again, it is deceiving to focus on iron in the upwelling regions when there are other regions that are iron limited.

OUR RESPONSE »>As stated in #2, we agree with this point and have changed the description of the tropical box as suggested.

5) L46: add "ppm" after approximately 46

OUR RESPONSE »>We changed this in the document.

6) L70: add "et al." after Durocher

OUR RESPONSE »>We changed this in the document.

7) L76-85: Remove iron-stressed and iron-limited. The sentence "Iron deposition in the tropical upwelling. . ." is not correct. There is actually very little iron deposition to the tropics, the North African dust plume deposits iron to the tropical Atlantic but that is the one example (see Jickells et al 2005 "Global Iron Connections Between

Desert Dust, Ocean Biogeochemistry, and Climate"). Iron is upwelled along the coasts in these areas along with macronutrients, but it is incorrect to call that iron deposition.

OUR RESPONSE »>We have removed iron-stressed and iron-limited from this section. Iron limitation will now only be referenced in the discussion.

Within our paper we have added the following sentence to address this comment, "Iron limitation is commonly thought to control [P] in the tropical upwelling regions (Moore et al., 2004; Raven and Falkowski, 1999) and the degree of nutrient drawdown has a strong impact on predicted (and observed) C:P in phytoplankton. This environmental control on C:P leads to highly non-linear controls on pCO2,atm whereby increased export in the tropics leads to increasing pCO2,atm."

8) L111-115: Questions 1 and 2 seem redundant, please remove question 1 or give more detail if it is in fact different from question 2.

OUR RESPONSE »>We recognize the confusion seen between the two research questions. The first is to determine the influence of cellular allocation strategies based on different environmental conditions (nutrients, temperature, and multi-environmental) on stoichiometric ratios. The second is to determine the influence of changing environmental conditions such as phosphorus concentrations and temperature on each stoichiometric model. In order to address this confusion, we have clarified the first question to include cellular allocation strategies.

Within our paper we have changed the research questions to read as follows: "We will explicitly address the following research questions: (1) How does environmental variability influence marine phytoplankton cellular allocation strategies and in turn the stoichiometric ratio? (2) What are the effects of changing environmental conditions on stoichiometric ratios, carbon export, and pCO2,atm?, and (3) What is the influence of the environmental gradients among the three major surface biomes on carbon export and pCO2,atm?"

9) L135: Where in the water column are you taking the phosphorus concentration?

OUR RESPONSE »> Phosphorus concentrations are prescribed within each box and then the model is run to steady state. The tropical and subtropical surface boxes extend down to a depth of 100 m, the high latitude surface box extends down to1000 m, the thermocline box extends from a depth of 100 m down to a depth of 1000 m, and the deep box extends down to a depth of 4000 m. For the use of phosphorus within our multi-environmental stoichiometric model we use the concentration in the respective surface box.

10) L207: add "et al." after Daines

OUR RESPONSE »>We changed this in the document.

11) L212: add "et al." after Daines

OUR RESPONSE »>We changed this in the document.

12) L213: Expanding on the phospholipid justification, can you explain more why you choose a zero contribution for phospholipids? Although non-P substitutes can reduce the phosphorus incorporated into P-lipids, observations suggest non-zero quantities remain. For example, 1.3 +/- 0.6% P uptake in the P-limited Sargasso are incorporated into phospholipids (Van Mooy et al 2009 – mentioned in next correction) and phospholipids make up approximately 5% of particulate organic P in the P-limited eastern Mediterranean (Popendorf et al 2011 "Gradients in intact polar diacylglycerolipids across the Mediterranean Sea are related to phosphate availability"). Might it be more appropriate to have two distinct P-lipid/total cellular P values for high and low phosphorus regions?

OUR RESPONSE»>We do in no way intend to imply that cells do not include P-lipids. Please see #3 for a detailed response to this point.

13) L215-216: For phospholipid substitution, a more appropriate reference would be Van Mooy et al 2009 "Phytoplankton in the ocean use non-phosphorus lipids in response to phosphorus scarcity" instead of Van Mooy et al 2006.

OUR RESPONSE ›>We have added the Van Mooy et al. 2009 reference.

14) L228: add "et al." after Daines

OUR RESPONSE ›>We changed this in the document.

15) L282: . . .that underlies the subtropical gyres and equatorial upwelling regions (labeled M), and deep waters. . .

OUR RESPONSE ›>We changed this is in the document.

16) L314: "Iron limitation is implicitly simulated through its control on the tropical [P]. . ." – how does iron control phosphorus concentrations? This is not clear in the manuscript and I personally have not come across any such research stating such. Again, if you are going to keep iron in the manuscript please provide references of where you have gotten the information and expand on the explanation of how you can make this justification.

OUR RESPONSE ›>Iron was removed from the manuscript and only discussed briefly in the discussion section.

17) Table 2: Please switch the columns so that Range of fhd (sv) is first and references is second (will be consistent with Table 1).

OUR RESPONSE ›>We have switched the column to be consistent with Table 1.

18) L350: "This set of experimental runs was intended to capture the effects of changing levels of iron deposition. . ." – Again, talking about iron deposition in these tropical up- welling regions does not make sense and as you have not provided references I would just remove it all together. This experiment wanted to test the sensitivity of pCO2 to nutrient availability, I believe that is a good enough reason and there is no need to mention iron limitation.

OUR RESPONSE »>We agree with this reviewer that and have removed this reference to Fe limitation.

19) L377: Change variables to variable

OUR RESPONSE »>We changed this in the document.

20) L379: Remove "iron stressed"

OUR RESPONSE »>We changed this in the document.

21) Figures 6 and 10: I really like these figures and think you could include more in the discussion about the implications of how global temperatures will affect export. It is a nice way to tie your work with large scale impacts on biogeochemical cycles and reiterate the importance of the study.

OUR RESPONSE »>We completely agree this with observations. We hope to expand on the potential implications of global temperatures effect on export based on findings.

22) L477: add "the" before data

OUR RESPONSE »>We changed this in the document.

23) L589: remove iron-limited

OUR RESPONSE »>We changed this in the document.

24) L596: remove "iron deposition or"

OUR RESPONSE »>We changed this in the document.

25) L600: remove sentence "This observation suggests that pCO2 may have a complex link: : :". You honestly have not shown anything to do with iron delivery and its link to pCO2, there is nothing included in the model that I saw and again have provided zero references about iron deposition

OUR RESPONSE »>We agree with the reviewer, it has been removed from the document. Instead, we linked it to macronutrient availability.

26) L650: remove "thus"

OUR RESPONSE »>We changed this in the document.

27) L674: remove "which might be influenced by increased atmospheric iron deposition,"

OUR RESPONSE »>We changed this in the document.

28) L680: change separating to separate

OUR RESPONSE »>We changed this in the document.

29) References: There are a few references that are not mentioned in the manuscript. A couple are about iron cycling and I am curious if and where they were originally included and also possibly had more of an explanation associated with them of why you link iron to phosphorus? Cunningham and John 2017 Moore 2004 Raven and Falkowski 1999 Also, please move Van Bogelen and Neidhardt 1990 and Van Mooy et al 2008 references to after the Toseland et al 2013 reference

OUR RESPONSE »>We apologize for the missing use of these references. This has been fixed in the manuscript.
* * *

---

## Author Comment (AC2) · 24 Dec 2017

We gratefully thank Referee #2 for their time, constructive comments, and suggestions to our manuscript. Below we have a detailed response to each comment posed by Referee #2. We have amended the manuscript in hopes that it will be much improved and our study presented clearer.

Anonymous Referee #2

Using a "classical" ocean carbon cycle box model and parameterizations of flexible elemental stoichiometry in surface ocean particulate, the authors examine the role of

elemental composition in mediating the response of atmospheric CO2 to ocean temperature change. There is a significantly different sensitivity when the particulate C:P ratio is represented by a "multi-environmental" (Ecological Stoichiometry and temperature dependent) framework compared to a fixed, Redfieldian particulate composition. Most notably, compensation between temperature sensitivity of solubility and biological pumps reduces the sensitivity to subtropical temperature change as well as reversing and enhancing the response to tropical perturbations. Variable elemental composition and phosphorus storage modify the sensitivity of atmospheric pCO2 to the efficiency of phosphate utilization in subtropics and tropics. The key message for is that variable elemental ratios have non-negligible impacts on the ocean's control of atmospheric CO2 and that the temperature sensitivity of solubility and stoichiometrically-mediated biological pumps have some interesting regional dependences.

I enjoyed reading this paper. I found it stimulating and thought provoking. There is a lot going on. The authors combine classical carbon cycle box model with several parameterizations of elemental stoichiometry of the sinking particulate (and/or primary producers). The authors connect cellular scale physiology and global carbon cycle and have used and developed an appropriate framework with which to do so.

My criticism of the paper is that the multi-environmental model is presented very much at face value. The assumptions and construction seem very logical but the choice and constraint of the parameters is by and large opaque. In particular the relationship between the storage component of the multi-environmental model and the Galbraith and Martiny parameterization seems interesting and important but is not really discussed. How important is the storage term in controlling the overall response of the multi-environmental model? It is not at all clear from the manuscript. I feel that some clarification and discussion along these lines is important for the reader.

I found the manuscript very interesting and thought provoking. I had a number of questions, comments and need clarification on certain points which I will detail here. Some are more important than others. While my recommendation is major revision, it

is clarification that I would like to see, not changes to what has been done.

30) The Sarmiento and Toggweiler and contemporary carbon cycle models focused a lot on the sensitivity of atmospheric pCO2 to "high latitude" changes. This isn't discussed here - perhaps there wasn't any as configured? Some comment would be useful and interesting in this regard.

OUR RESPONSE ›› In this manuscript, we are focusing our efforts on the potential impacts of the low latitudes. Sarmiento and Toggweiler both found that high latitudes have a big impact on atmospheric pCO2. In no way do we disagree with this seminal work, we simply are trying to bring attention to the importance of low latitude processes as an additional mechanism(s) to consider when predicted biogeochemical feedbacks. We take their original findings into consideration when creating the model. In the box model, fhd, a bidirectional mixing term that ventilates the deep box directly through the high-latitude surface box, has a large impact on the magnitude of atmospheric pCO2. We prescribed the baseline value to be 45.6 Sv in our model but when we increase it to 108 Sv, the change in pCO2 is ∼105 ppm for C:P at Redfield proportions. Thus, high-latitude processes clearly have a major impact on ocean and global biogeochemistry. To address this comment in the manuscript, we have added the following: "Although the focus of this study is to determine the impact of low latitudes on pCO2,atm, we point out that at Redfield stoichiometry, pCO2,atm increases by 100 ppm when fhd is increased to 108 Sv from its default value 45.6 Sv.

31) The stoichiometry of sinking particulate and of primary producers is certainly connected but not necessarily the same. The multi-environmental model is founded on primary producer physiology. Perhaps this potential difference should be flagged?

OUR RESPONSE ››We acknowledge that sinking particulate stoichiometry and primary producer stoichiometry can be different in certain regions but overall, it has been found to be reasonably linked (Teng et al. 2014). Thus, we find that this assumption is reasonable to a first order.

Within our paper we have added the following statement, "For certain values of the parameters, the model produced excessive nutrient trapping in the thermocline. In order to dampen the nutrient trapping, we tuned the remineralization depth. Assuming that 25% of the total export is respired in the thermocline with the remaining 75% exported into the deep ocean, produced a better match between the modeled and observed [P] in the thermocline box. Total export is made from both the stoichiometry of sinking particulate and of primary producers, based on Teng et al. (2014) this is a reasonable first order assumption."

32) I would have been interested to see a Droop-style model in the mix as its a relatively common tool - but there is more than enough going on here anyway.

OUR RESPONSE »>We believe that the Galbraith and Martiny (2015) model (nutrient-only model in our study) is qualitatively similar to the Droop model. Thus, we expect the outcome to be very similar (i.e., a direct dependence of C:P on P availability).

33) I very much like the spirit of the multi-environmental model. (Though I find the name a little odd). It accounts for the role of cell size in mediating nutrient affinity and cell composition (contribution of cell wall material). It was not made clear how sensitive the final parameterization or the outcomes of the box model are to the assumed cell size. Nor could I find any information about the cell size assumed (or modeled?) in the simulations.

OUR RESPONSE »>We do not assume that there is a single size characterizing all phytoplankton cells in our model. Instead, cell size is one of the key elements of cell strategy that we model. Smaller cells have greater specific nutrient uptake rates, but their cell wall and membrane occupies a greater fraction of their biomass than larger cells, and thus they have less space (specific to biomass) for investments in either photosynthesis or biosynthesis.

Figure 4 and 5 are meant to illustrate the predictions that our model makes in different environmental conditions. In the original manuscript, these figures showed model

predictions for C:P ratios, biosynthesis investment, and photosynthesis investments. However, in each of these figures, the predicted cell radius is also varying. In order to make the predictions of our model clearer, we have augmented each of Figure 4 and 5 with an additional plot, showing how cell radius varies with environmental conditions.

Our model predicts a strong relationship between nutrient concentrations and cell size. In oligotrophic conditions the model predicts a radius under 1 $\mu$m. When resources are abundant the model predicts much larger cells. Our model also predicts a weak, but non-zero dependence of both irradiance and temperature on cell size. Higher irradiances lead to smaller cells (due to a lower requirement for photosynthetic machinery), and there is a non-monotonic, concave relationship between temperature and cell size, which is due to a subtle interaction between biosynthesis efficiency (which varies greatly with T) and size dependent uptake rates.

34) The photosynthesis parameterization and allocation scheme is very reminiscent of Geider's models in spirit and mathematical form. What is the relationship?

OUR RESPONSE »>Our model is very closely related to the multi-compartment photosynthesis model presented in Talmy et al. 2013 (we incorrectly cited Talmy 2014 in the model description, this has been changed to correctly cite this paper). Geider is a co-author of this paper and indeed the modeling framework presented there is very much consistent with the photosynthetic models he has devised throughout his career. We utilized the functional responses which they derived in that paper to represent the allocation of photosynthetic machinery to either light harvesting or carbon fixation. Their model included other compartments (photoprotection and biosynthesis) which were suited to the particular dynamic light environment that they were interested in studying. We use our own parametrization for biosynthesis. Talmy et al. 2013 found that the photoprotection allocation was not a large or greatly changing component of their allocations. We have therefore not included it would complicate our model with little change in our qualitative results.

On the other hand, the decomposition of photosynthesis into light harvesting and carbon fixation components is critical, and makes our model predictions agree much better with experiments studying the variations of C:P or N:P ratios with irradiance. Models that do not have this decomposition predict too large of a decrease in cellular allocations to photosynthesis at high-light levels. In a two compartment model, increases in allocations to carbon fixation cause the overall allocation to light harvesting to have a more mild decrease. The two-compartment treatment also seems more physiologically realistic than a 1-compartment, which only models photosynthetic pigments. Thus we used the functional forms and parameters that were derived (experimentally) in Talmy et al. 2013 for carbon-fixation and light-harvesting. We have added a small amount of text to better clarify the relationship.

35) The statement at line 235 that the "unique maximum of the growth rate occurs for the set of parameters that lead to co-limitation by nutrients, photosynthesis and biosynthesis" is very interesting and intriguing. Is that an emergent property? Is it obvious that it should be this way? I would have liked to hear more about this.

OUR RESPONSE »>It is commonly the case for mathematical models like ours that model the tradeoffs between different allocations of biomass to different physiological functions to have a unique solution with a maximum growth rate. The reason is that if one increases the investment a cell makes in some pool, this will decrease the investments in other pools. Thus, the only way for a cell to increase the photosynthetic rate is to decrease either biosynthesis or nutrient uptake rates, and vice versa. In such cases we generally find a unique solution at which the three rates, $\mu$Photo, $\mu$E, and phosphorus are the same. This might be obvious for very simple models, or for people who primarily work using these models. Indeed, the easiest way to see what is going on is to imagine a simpler model with a similar style. For example, if we modeled a cell with fixed radius which is limited by either light or by biosynthesis, then the cell growth rate would be min($\mu$Photo, $\mu$E). If we start with the biosynthesis allocation at 0, the growth rate of the cell will be $\mu$E = 0, but $\mu$Photo will be high because of all of

the photosynthesis proteins. As E increases, $\mu E$ increases, and the cell grows faster. At this time, $\mu Photo$ is going down, but this doesn't affect mu. At some value of E, $\mu E$ = $\mu Photo$ (since if E = 1, $\mu Photo$ = 0). This point will be the optimal strategy, since further increases in E will cause a switch to limitation by $\mu Photo$, which decreases with increasing E.

We have actually been able to convert the intuitive picture described above into a mathematical proof. Since it does not require much additional space to include it, we have added it to the paper, along with a figure indicating graphically the idea behind the proof.

36) Phosphorus storage seems to be very important. Equation (13) controls a residual storage pool that constrains the parameterized stoichiometry to match the observed relationship between phosphate and particulate stoichiometry, as I understand it. Thus it strongly mirrors the Galbraith and Martiny model of equation (1). For me, some key questions concern this aspect of the model: How significant in the overall control of model stoichiometry is this component? If it dominates, then I could view the multi-environmental model in some way as a combination of the Galbraith and Martiny model with temperature sensitivity. Or does the more mechanistic and detailed physiology have a significant role? Either way, I think the mechanistic model is valuable and interesting but I would like to understand how much the results are driven by the storage of phosphorus. Its important for a number of reasons and I feel that this should be clearly discussed.

OUR RESPONSE »>The impact of the residual pool on the overall size of the P pool is heavily dependent on environmental conditions, varying from a minimum of close to 0% to a maximum of just under 50%, for the combinations of parameter values used in all of our numerical experiments. Over most of the parameter range considered here, the contribution of the residual pool is much more modest, 10-20%. High values occur when phosphorus is available and the temperature is high. In these conditions, ribosomal contributions are decreased, but the residual contribution is high. In cold water,

high P ecosystems, the residual contribution is approximately 25%, and in oligotrophic ecosystems it is close to 0.

Thus, we view the mechanistic/physiological part of the model as being more significant, but it is important to acknowledge that we don't believe that this mechanistic model can on its own explain all of the observations of C:P in the ocean. In particular, it is not possible for the purely mechanistic model to predict the extremely low C:P ratios observed in some ocean regions. This is because the C:P ratio of the biosynthesis apparatus sets a lower limit. Even if we assume that proteins made an even smaller contribution to biosynthesis (which would cause biosynthesis to have a lower C:P), it would still be impossible to match the most extreme observations. If the C:P of the biosynthesis pool was variable or lower, then the contribution of residual pool would be somewhat smaller, but still necessary. (Better understanding the balance between ribosomes and non-photosynthetic proteins is likely a good direction for future research).

In order to make the importance of storage, we have added an additional plot reveals the relative contribution of the storage pool to the total P pool as a function of environmental conditions, with a short discussion.

37) A small thing, but I had to stop and think about equation (1) because [P]o has different dimensions than [P]: the former is a ratio and the latter a concentration. I think it would be much clearer and more appropriate to denote [P]o as in (13), with a symbol in accord with other variables that are ratios.

OUR RESPONSE »>Thank you for noticing this, we have changed the notation so that [P]0 is now (P:C) 0, i.e. the P:C ratio predicted by linear regression at zero P concentration.

38) The box model formulation makes sense; the inclusion of the thermocline reservoir is important for the sensitivity to changes in the subtropical surface. Some small details: how is the carbonate system solved? Is alkalinity fixed or is there an implicit carbonate

pump?

OUR RESPONSE »>Thank you for bringing this to our attention. The nonlinear carbonate system equations are solved using Matlab's fsolve function. We calculated the solubility constants using biome specific salinity and temperature. The solubility constant then is used to break up total carbon into $pCO_2$, bicarbonate and carbonate ions. Total carbon is quantified using the breakdown of carbon ions ($pCO_2$, bicarbonate and carbonate) and alkalinity concentration. Total carbon and its breakdowns (which we keep track of at each time step) are transported laterally to each box through our thermohaline circulation. We have added more detail to our box model design description, to make sure we are clear on how this model is created.

Within our paper we have added these lines to address this comment: "To quantify the breakdown of carbon into these components, we model the solubility pump, using temperature and salinity to determine the partitioning of inorganic carbon among total carbon within a box. The global mean alkalinity is prescribed according to the observed mean ocean values. Our box model simulations various forms of C similar to alkalinity. Biome specific salinity and temperature are used to prescribe the solubility constants of $CO_2$ in seawater and the bromine concentration, which is taken to be proportional to salinity. We use these calculations to determine the $pCO_2$ value at standard pressure (1 atm) within each box. Box specific total carbon is calculated from the $pCO_2$ value, bicarbonate, carbonate and alkalinity concentrations. $CO_2$ cycles through the atmosphere via the air-sea gas exchange fluxes (fah, fas, fat). We used a uniform piston velocity of $5.5 \times 10\text{-}5$ m s-1 to drive air-sea gas exchange (DeVries & Primeau 2009, Follows et al. 2002)."

39) The model doesn't resolve nitrogen, and I would expect that the allocation of nitrogen in proteins and pigments would be an important factor, perhaps more so than phosphorus. Does this actually matter? A comment on this would be helpful.

OUR RESPONSE »>This is an important point that was also raised by reviewer 1(Referee 1: Response #1). In short, the reason for using P is its role as the ultimate limiting nutrient on long time-scales as well as to simplify the model and avoid an explicit N cycle.

40) Why does temperature affect biosynthesis but not photosynthesis (line 398) imposed from empirical observations? The model description tells us that Q10=2 for temperature dependences, but there is no discussion with regard to photosynthesis. Why this choice?

OUR RESPONSE »>We model photosynthesis as having a Q10=1, which is consistent with physiological studies going back to Shuter 1979 that suggest that photosynthetic efficiency does not depend on temperature over physiologically relevant ranges. The discrepancy between photosynthetic and biosynthetic temperature dependence has traditionally been explained by referring to the differences in the chemistry and physics of the two processes. The electron transport chain relies on quantum mechanical processes, which are unaffected by variations in temperature in a physiologically relevant range. A good reference for this is Devault 1980, Quantum Mechanical Tunneling in Biological Systems. We have added some text to more explicitly explain our choice of temperature dependence parameters for different processes.

41) The discussion of sensitivity to cell radius in lines 400-410 doesn't tell us what is the cell radius (or distribution of) in the model? Is it imposed or modeled (I presume the former but nothing is said in the paper). This should be clear.

OUR RESPONSE »>Cell radius is an emerging property based on the phosphorus concentration, light, and temperature on the cell.

42) Figures 3,4,5 are a bit small and fuzzy when printed.

OUR RESPONSE »>We have fixed these figures so they are clearer.

43) I'd really like to understand how important the storage term is in the overall control of figures 4 and 5. We see the variation in C:P and the relative allocation to biosynthesis

and photosynthesis, but it's not clear how important the latter is to the former.

OUR RESPONSE »>We have included an additional figure as part of our response to an earlier comment about the importance of phosphorus storage.

44) What is the cell size in the box model simulations? Is it imposed? Does it vary? How sensitive are results to r?

OUR RESPONSE »>The cell size varies based on the phosphorus concentration, light, and temperature in each surface box when we are running the multi-environmental stoichiometry. When using the Redfield, nutrient-only and temperature-only stoichiometric models there are not explicit cell sizes but implicit varying ones.

45) The model is P based. However, as is alluded to in the manuscript, nitrogen and iron dynamics are important. Indeed P is found to be the proximal limiting in only a few areas of the global ocean, with N and Fe controlling things locally. So how does this affect the relevance of the model? Wouldn't N and Fe dynamics be more important at the individual scale? Would this (does this) mean that storage is most significant for P:C? Again, understanding the significance of storage for the outcomes here is very important.

OUR RESPONSE »>It is true that phosphorus rarely is a proximal limiting nutrient, whereas nitrogen and iron commonly limit productivity in the short term. However, on long time-scales P is commonly considered the ultimate limiting nutrient and our results are indeed based on long-term equilibrium states. However, it is also clear that the three nutrient cycles have complex interactions both within and outside the cell and we hope to add explicit N and Fe cycles in future iterations of the model.

46) Line 464: "nutirent"

OUR RESPONSE »>We have changed it in the document.

47) The contrasting temperature sensitivities of tropical and subtropical perturbations is very interesting. The dominance of the solubility term in subtropical responses is

ascribed to the "large surface area" of the subtropical region (line 563). I don't think that's true: I think its because the subtropical surface feeds the subtropical thermocline which represents a significant contribution to global water volume. Hence, changes in subtropical solubility have significant leverage. Since tropical waters don't directly feed into any subsurface water mass, they do not have the same leverage. This is why the resolution of the thermocline box is important. The classic Harvardton Bear box models did not resolve the thermocline and so found very low sensitivity to subtropical perturbations relative to 3D circulation models. Resolving the thermocline in the box models brings them into consistency (this was the point of Follows et al, 2002). I thought this was why the authors had chosen the configuration which resolves a thermocline reservoir.

OUR RESPONSE »> The reviewer is correct. The sensitivity of atmospheric CO2 to solubility changes in a box in contact with the atmosphere depends on the volume of the subsurface ocean ventilated from that box and on the degree of air-sea disequilibrium as explained in Follows et al, 2002 and also in DeVries and Primeau 2009. (The disequilibrium effect can be significant for high latitude boxes that have a relatively small surface area and a vigorous exchange rate with deeper water masses). We thank the reviewer for allowing us to clarify the point we were trying to make, which is that because the nutrient supply to the subtropical gyres is dominated by the lateral transport of unused nutrients from the tropical box rather than by vertical exchange, the strength of the biological pump does not scale with the surface area of the subtropical gyre, whereas the volume of the thermocline box very roughly speaking scales as the area of the subtropical box, at least in the limit where the surface area of the tropical box is negligible compare to that of the subtropical box, simply because volume is equal to area times thickness. To make the text clearer and more accurate we have revised it as follows:

The decrease in surface CO2 solubility at high temperatures is sufficient to overcome the increase in export due to higher C:P leading to a positive relationship between

[Figure]

pCO2,atm and subtropical temperatures. It is important to point out that the relative importance of the two competing effect depends critically on the physical circulation of the ocean. Predicted increases in stratification are often invoked as a mechanism that would decrease the vertical supply of nutrients, which one might think would further compensate for the effect of higher C:P. However, the strength of the biological pump in the subtropics is controlled the by lateral transport of nutrients rather than by vertical exchange so that the impact of increasing stratification might not be important. Similarly, it is unclear how increases in stratification might affect the strength of the solubility pump. The sensitivity of pCO2,atm to changes in subtropical surface temperatures depends critically on the volume of the ocean ventilated from the subtropics, i.e. on the volume of the thermocline box in our model. How this volume might change in response to a warming world is a complicated dynamical problem that is beyond the scope of the present work.

---

## Author Response (AR2)

We thank the Associate Editor and reviewers for their time in reading our manuscript and have a full detailed response to each comment. Lastly, we have our revised manuscript. We hope that our manuscript is strong, organized and well representative of our study.

5 Dear Authors,

While I feel that your manuscript has improved by addressing the previous reviewers comments, the third reviewer has raised a few additional points, mostly concerning the presentation that I would like you to consider.

Best regards,
Katja Fennel

OUR RESPONSE
15 >>> In order to address this commentary from Reviewer #3, we have switched the order of our second section to improve the overall organization and flow of the manuscript, and have edited Figure 8C and D to be easier for colorblind individuals to read. Additionally, we have included the five box model equations to the second section and posted the multi-environmental model on GitHub (https://github.com/georgehagstrom/-bg-2017-367-/blob/master/CP.m).
20 _______

We gratefully thank Referee #3 for their time, constructive comments, and suggestions to our manuscript. Below we have a detailed response to each comment posed by Referee #3. We have amended the manuscript in hopes that it will be much improved and our study presented clearer.

25 Anonymous Referee #3

This manuscript has already gone one round of review and revision. I generally agree with the comments by the previous reviewers. The comments below mostly concern the presentation of
30 this work, not the scientific substance.

1) A complete set of model equations is missing and should be added, especially for the 5-ocean box model framework and the new multi-environmental model. While many descriptions for the former and parameterizations of the latter are given in sections 2.1.4
35 and 2.2, none of the full model equations are given. This makes it really hard for the reader to grasp how the model actually works. I certainly don't feel I could reproduce what was done based on the information provided.

OUR RESPONSE
>>> In order to address this we have added the 5-ocean box model equations within the
40 model description section. We amended the model description to add in the equations as follows: "The conservation equations of phosphorus are as follows:

$$\frac{dP_T}{dt} = \frac{(P_M - P_T) \cdot Tc + (P_M - P_T) \cdot Tw - (a + b) \cdot Pt}{VT}$$

$$\frac{dP_S}{dt} = \frac{(P_T - P_S) \cdot Tc + (P_T - P_S) \cdot Tw - (c + d) \cdot Ps}{VS}$$

$$\frac{dP_H}{dt} = \frac{(P_S - P_H) \cdot Tc + (P_D - P_H) \cdot fhd - Ph}{VH}$$

$$\frac{dP_M}{dt} = \frac{(P_D - P_M) \cdot Tc + (P_S - P_M) \cdot Tw + a \cdot Pt + c \cdot Ps}{VM}$$

$$\frac{dP_D}{dt} = \frac{(P_H - P_D) \cdot Tc + (P_H - P_D) \cdot fhd + Ph + b \cdot Pt + d \cdot PS}{VD}$$

where P represents the concentration of phosphorus at a specific box, a and c represents 0.25 remineralization, b and d represents 0.75 remineralization, and V represents the volume of the specified box."

In terms of the multi-environmental model, our description does have many of the parameterizations of the model. As such we have placed the full model equations on GitHub (https://github.com/georgehagstrom/-bg-2017-367-/blob/master/CP.m) for those interested in recreating the model.

2) It would also be more logical to me if the order of the first part of section 2.2 (the text describing the box model) and section 2.1 was switched. The current section 2.2.1 could follow on from the current 2.1.

OUR RESPONSE

>>>We agree that switching the model description sections provides a better flow in the manuscript. As such, we have moved section 2.2 to be first followed by section 2.1 and 2.2.1 (now 2.3).

3) Similar to Reviewer 1, I'm concerned about the complete omission of N in the model. I'm not sure the response adequately addresses this concern. I would be more comfortable if the authors acknowledged that variability in N could affect the results. They seem to do so in the response, but not in the modified manuscript.

OUR RESPONSE

>>> This is a very important point. As stated previously, readers need to understand our reasoning for omitting N in the model. To make this point clear and more explicit we have added more information to our choice of omitting N.

We have amended the manuscript to better represent this change in the following way:

"Phosphorus is used to represent the role of nutrient availability in controlling stoichiometry and C export. We chose this over N because on long time-scales, P is commonly considered the ultimate limiting nutrient whereas N is only limiting productivity and export on short time-scales (Tyrrell, 1999). On long time-scales, nitrogen fixation/denitrification will presumably adjust the N inventory. Our modeling is focused on long term steady-state outcomes and we would like to avoid issues associated with modeling the N cycle (like getting N-fixation and denitrification rates correct). Thus, we chose to use P as a

representative for nutrient availability representing the long-term steady-state biogeochemical equilibrium."

85 Minor comments:
4) Line 54: "flu" should be changed to "flux"
OUR RESPONSE
>>> We changed this in the document.

90 5) The color choices in the color Figures, especially in Fig 8 C and D are not good for colorblind readers. Suggest the authors pick something better.
OUR RESPONSE
>>> Thank you for the suggestion. We have modified Figure 8C and D so that it is easier for color blinded individuals.
95 _________

[revised manuscript text omitted]

al.(1979) or Daines  et al.(2014)(Daines et al., 2014)(Daines et al., 2014)(Daines et al., 2014)(Daines et al., 2014), who assume that uptake rates are diffusion-limited. (2014), who assume that uptake rates are diffusion-limited.

440    The phosphorus quota for functional elements of the cell (thus not including any storage) is determined by the allocation to biosynthesis $E$ and the percentage $p_{DNA}$ of cellular dry mass allocated to DNA:

Formatted Table

$$Q_{p,biosynthesis}(E,r)Q_p(E,r) = \frac{4}{3}\pi r^3 \rho_{\text{cell}} p_{\text{dry}} \frac{(\alpha_E E P_{\text{rib}} + p_{\text{DNA}} P_{\text{DNA}})}{31}. \qquad (148)$$

Here, we assume that there is no contribution to the functional-apparatus P quota from
445 phospholipids, which instead is are merged with storage molecules. This differs from Daines et al. (2014).(2014), who assumes that phospholipids occupy 10% of the cell by mass. Phytoplankton can substitute sulfoquinovosdiaglycerol (SQDG) for phospholipids in their cell membranes under low P conditions (Van Mooy et al., 2009). Similarly, P storage molecules are also regulated by P availability. Thus, we here assume that treat
450 phospholipids and P-storage exhibit the same behavior and thus model-wise treated as one pool. Phytoplankton can substitute sulfoquinovosdiaglycerol (SQDG) for phospholipids in their cell membranes in low P conditions ((Van Mooy et al., 2009)
The function $f_P$ is the cellular. , implying that it is appropriate to functionally treat them together with the storage pool.
455    The function $f_P$ is the response of the cell to light levels, and is chosen to capture the effects of both electron transport and carbon fixation on photosynthesis, and is closely related to a previous model derived by Geider and Talmy(Talmy et al., 2013)(Talmy et al., 2014)(Talmy et al., 2014)(Talmy et al., 2014)(Talmy et al., 2014). This prior model included four compartments: electron transport, carbon fixation, photoprotection, and
460 biosynthesis. Talmy and co-workers 4)3(Talmy et al., 201. Their model included four compartments: electron transport, carbon fixation, photoprotection, and biosynthesis. Talmy (2013) foundIt was(2013) found that the photoprotection allocation ,was not a large or greatly changing component of their allocations. We therefore do not include this within our model due to its high complexity with little qualitative results. Our biosynthesis was
465 also separately parametrized. We also separately parametrized biosynthesis.it because it would require complicate our model with little change in our qualitative results. Including We have therefore not included.

[revised manuscript text omitted]
 temperatureT of 26°C, box S has a temperatureT of 24°C, and box H has a temperatureT of 7°C. Box S covers 39% and Box T covers 25% of the ocean surface area.*

To simulate the global transport of water between boxes, our model includes a thermohaline circulation (labeled Tc) that upwells water from the deep ocean into the tropics, laterally transports water into the subtropics and high-latitudes, and downwells water from the high-latitude region to the deep ocean. Surface winds produce a shallow overturning circulation (labeled Tw), that transports water from the thermocline to the tropics and then laterally into the subtropics. These circulations create teleconnections of nutrient supply in the surface ocean boxes. A bidirectional mixing term that ventilates the deep box directly through the high-latitude surface box (labeled fhd) represents deep water formation in the Northern Atlantic region and around Antarctica (S(Sarmiento and Toggweiler, 1984). The parameters Tc, Tw and fhd are considered adjustable parameters, which we calibrate using phosphate data from WOA13 (G(Garcia et al., 2014). In order to simulate the movement of particles, we included export fluxes (Pt, Ps, and Ph) of organic phosphorus out of each surface box.

Our box model simulates [P], alkalinity and various forms of C; total carbon in the surface boxes is partitioned into carbonate, bicarbonate, and $pCO_2$. The global mean [P] is prescribed according to the observed mean ocean value (G(Garcia et al., 2014). The export of carbon is linked to phosphorus export using the C:P$_{export}$ ratio. but is also subject to transport (Sarmiento and Toggweiler, 1984)simulatesatesationsvarious forms of C similar to alkalinity alkalinity and total inorganic carbon, which are conserved tracers from which the speciation of inorganic carbon in sea-water can be calculatedWe use these calculations to determine the $pCO_2$ value at standard pressure (1 atm) within each box. Box specific

total carbon is calculated from the $pCO_2$ value, bicarbonate, carbonate and alkalinity concentrations. ((DeVries and Primeau, 2009; Follows et al., 2002). 
[revised manuscript text omitted]